
# Long-term airborne measurements of pollutants over the UK, including during the COVID-19 pandemic, to support air quality model development and evaluation

Angela Mynard[1], Joss Kent[1], Eleanor R Smith[1], Andy Wilson[1], Kirsty Wivell[1], Noel Nelson[1], Matthew Hort[1], James Bowles[1], David Tiddeman[1], Justin M. Langridge[1], Benjamin Drummond[1] and Steven J. Abel[1]

[1]Met Office, Exeter, Devon, EX3 1PB, UK

*Correspondence to*: Angela Mynard (angela.mynard@metoffice.gov.uk)

**Abstract**

The ability of regional air quality models to skilfully represent pollutant distributions throughout the atmospheric column is important to enabling their skilful prediction at the surface. This provides a requirement for model evaluation at elevated altitudes, though observation datasets available for this purpose are limited. This is particularly true of those offering sampling over extended time periods. To address this requirement and support evaluation of regional air quality models such as the UK Met Offices Air Quality in the Unified Model (AQUM), a long-term, quality assured, dataset of the three-dimensional distribution of key pollutants has been collected over the southern United Kingdom from June 2019 to April 2022. This sampling period encompasses operations during the global COVID-19 pandemic, and as such the dataset serves an additional application in providing a unique resource with which to explore changes in atmospheric composition associated with reduced emissions during this period. Measurements were collected using the Met Office Atmospheric Survey Aircraft (MOASA), a Cessna-421 instrumented for this project to measure gaseous nitrogen dioxide, ozone, sulphur dioxide and fine mode ($PM_{2.5}$) aerosol. This paper provides a technical introduction to the MOASA measurement platform, flight strategies and instrumentation. The MOASA air quality dataset includes 63 flight sorties (totalling over 150 hours of sampling), the data from which are openly available for use. Example case studies using data from these sorties are presented, which include an analysis of the spatial scales of measured pollutant variability, initial work to evaluate performance of the AQUM regional air quality model, and an introduction to the vertical structure of pollutants observed during repeated flight patterns over Greater London, including during the COVID-19 impacted period.

## 1 Introduction

The World Health Organisation identifies atmospheric air pollution as the single largest environmental risk to human health globally (World Health Organization (WHO), 2017). Long-term exposure to anthropogenic air pollution is linked with increased morbidity rates and premature mortality from chronic diseases (Air Quality Expert Group, 2020, Manisalidis et al., 2020) , which in the UK alone is estimated to have an annual impact on shortening lifespans equivalent to 28 – 36 thousand deaths (DEFRA, 2019). The impacts of air pollution on human health can be most acute in urban areas, particularly megacities, where high pollutant concentrations



coincide with high population densities (Molina and Molina, 2004). In addition to impacting human health, air pollution has been shown to have wider detrimental impacts on ecosystems, including animal welfare, crop yields, waterways, biodiversity and visibility (DEFRA, 2019).

From an atmospheric sciences perspective, air pollution is a complex, transboundary problem. Gaseous and particulate pollutants originate from many sources, are subject to transport and mixing over a range of scales

and undergo complex physical and chemical processing prior to deposition. In order to develop effective strategies for mitigating the impacts of air pollution, for example through emission control and limiting population exposure, these processes must be understood and leveraged to provide predictive capability extending spatially and temporally beyond the ground-truth provided by observations. Atmospheric chemical transport models represent a key tool in this domain.

Air quality models vary widely in spatial scale and complexity and have evolved rapidly in sophistication in recent years. The reader is directed to El-Harbawi (2013) for a comprehensive review of air quality modelling systems, that span scales from street canyon to global and incorporate a wide range of schemes representing pollutant emissions, turbulent mixing, advection, gas-phase chemistry and aerosol processes. Many of these models run online, meaning meteorological and pollutant fields evolve prognostically within the modelling

system allowing feedbacks between the two to be represented (such as direct and indirect aerosol effects) (Savage et al., 2013).

In the Met Office, the primary air quality modelling system is the Air Quality in the Unified Model, AQUM, a limited area forecast configuration of the Met Office Unified Model (MetUM) (Savage et al., 2013, Walters et al., 2019). AQUM provides daily UK national air quality forecasts of the Daily Air Quality Index (DAQI) up to

five days ahead (see https://uk-air.defra.gov.uk/forecasting/). The DAQI is generated from the forecast of nitrogen dioxide ($NO_2$), sulphur dioxide ($SO_2$), ozone ($O_3$) and particulate matter (diameters ($D_p$) <2.5 μm: $PM_{2.5}$ and $D_p$ <10 μm: $PM_{10}$) concentrations. The AQUM 12 km horizontal resolution grid covers much of western Europe (Savage et al., 2013) with 63 vertical levels up to a top height of 39km, where levels are non-uniform and the vertical resolution becomes coarser away from the surface. AQUM derives its boundary

conditions from the MetUM global forecast model (meteorological fields) and GEMS/MACC global models (chemistry and aerosol fields) (Flemming et al., 2009). Within the model domain, emissions over the UK are derived from the UK National Atmospheric Emissions Inventory (NAEI, (Thistlethwaite et al., 2013)), which has a resolution of 1 km. Atmospheric chemistry is represented by the UK Chemistry and Aerosols (UKCA) Regional Air Quality chemistry scheme (OConnor et al., 2014), and aerosol processes by the Coupled

Largescale Aerosol Simulator for Studies in Climate (CLASSIC) scheme (Bellouin et al., 2011). Given the resolution of AQUM, it is best suited to modelling background and regional air quality away from strong, very localised sources of pollution. A comprehensive description of the AQUM is available in Savage et al. (2013).

Air quality models, including AQUM, require high quality observations for development and evaluation. Given that air quality regulatory limits are imposed at ground level only, air quality model evaluation studies typically

focus on assessment of performance using surface measurements. In the UK, these observations are commonly provided by the Automatic Urban and Rural Network (AURN), an automatic ground monitoring network operated on behalf of the UK Department of Environment, Food and Rural Affairs (DEFRA) (Yardley et al.,



2012). AURN consists of around 70 sites in rural, remote, urban background and suburban settings, providing hourly measurements of nitrogen oxides ($NO_x$), sulphur dioxide ($SO_2$), ozone ($O_3$), carbon monoxide (CO), fine particulate matter ($PM_{2.5}$) and coarse particulate matter ($PM_{10}$) (Yardley et al., 2012), although not all species are measured at all sites.

In a comparison of AQUM to AURN observations, Savage et al. (2013) found that AQUM generally performed well, in particular for large air quality events, but had a number of systematic biases. For example, a positive bias in ozone at urban sites, a positive $NO_x$ bias at rural sites and a negative bias at urban sites and general negative biases in both $PM_{2.5}$ and $PM_{10}$. Ground based observations are used to bias-correct the model data and minimise some of these systematic biases at the surface (Neal et al., 2014). We note that these biases may not solely be due to model performance and could also be partially attributable to difficulties in evaluating a 12 km resolution model with point observations that have limited spatial coverage, both in the horizontal (raising questions of representivity) and in the vertical (limiting model evaluation away from the surface-atmosphere boundary). These limitations in observational data currently available for model evaluation provide motivation for the current work, with a particular focus on the need for observations away from the surface. Given that vertical mixing serves to transport pollutants both away-from and towards the surface, and pollutant chemical, physical and removal processes occur throughout the atmospheric column, model skill in this domain is critical to achieving successful prediction at the surface (Solazzo et al., 2013).

Observations of pollutants throughout the atmospheric column are increasingly available from satellite instruments (e.g. Tropomi on ESAs Sentinel-5P (Veefkind et al., 2012, Air Quality Expert Group, 2020, Wyche et al., 2021) and GOME on ESAs ERS-2 (Molina and Molina, 2004)). While these observations can provide global coverage extending over timescales of years, they generally contain limited information on the vertical distribution of pollutants within the column (Fleming, 1996, Peers et al., 2019). Instrumented aircraft provide one way of addressing this gap. Over several decades, there have been a number of related large-scale initiatives to instrument in-service commercial aircraft to provide such measurements, for example Measurements of OZone, water vapour, carbon monoxide and nitrogen oxides by Airbus In-service airCraft (MOZAIC, Solazzo et al., 2013) and In-service Aircraft for a Global Observing System (IAGOS, (Petzold et al., 2015)). Over forty-four thousand flights have been conducted under IAGOS since 1994 and though temporally and spatially restricted by commercial flight patterns and timings, these projects serve as a prime example of the use of instrumented aircraft to provide long term observations for atmospheric model evaluation. An alternative approach is the use of atmospheric research aircraft (ARA), which are aircraft instrumented and deployed specifically for the pursuit of atmospheric science and monitoring. ARA deployments tend to focus on specific locations or events and instrument payloads can vary greatly dependent on the phenomenon under study. As such, while ARA are particularly well suited to the detailed study of chemical and physical processes (a key requirement for model development), the often-sporadic nature of their deployment limits the generation of consistent, long-term datasets. It is this gap that this work seeks to fill with a specific focus on air quality observations over the UK to allow for the evaluation of regional models such as AQUM.



The UK Clean Air: Analysis and Solutions research programme is led by the Met Office and Natural Environment Research Council (NERC) and has invested in modelling, data and analytical tools to assess current and future air quality and the impact of policies designed to improve it (DEFRA, 2019). Under this umbrella, a long-term, quality assured dataset of the three-dimensional distribution of key pollutants ($NO_2$, $O_3$, $SO_2$ and $PM_{2.5}$) has been collected using the instrumented Met Office Atmospheric Survey Aircraft (MOASA). Observations have primarily covered the southern UK, including Greater London, with 63 flights throughout the period 2019-2022. This paper introduces the strategy and quality assurance basis for these observations, with the intention of serving as a comprehensive technical reference for all future users of these data. In particular it includes descriptions of: i) the measurement platform and instrumentation, ii) flight strategies, iii) analysis of the spatial scales of measured pollutant variability, iv) initial use of these data to evaluate performance of the AQUM regional air quality model, and v) an introduction to the vertical structure of pollutants observed during repeated flight patterns conducted over Greater London during the COVID-19 impacted period.

### 1.1 Impact of COVID-19

In January 2020, the first case of severe acute respiratory syndrome coronavirus (SARS-CoV-2), referred to as COVID-19, was identified in the UK (Jephcote et al., 2020). Since 24[th] March 2020, to curtail person-to-person transmission of the virus, the United Kingdom has been subject to various levels of lawful regulation limiting all non-essential travel and contact. A consequence of the restrictions has been a reduction in mobility (50-75% across major cities during the Spring 2020 lockdown, Air Quality Expert Group, 2020) as businesses switched to homeworking, and industry and commercial sectors reduced operations. This resulted in a significant drop in emissions of primary air pollutants, most markedly from the transport sector (road, rail, and aviation) and in urban environments. Similar impacts have been seen across Europe (Lee et al., 2020) and have collectively resulted in significant changes to UK air quality compared to the climatological norm (Air Quality Expert Group, 2020).

Flight operations with the MOASA aircraft encompass periods in 2020 and 2021 impacted by these COVID-related changes to air quality over the UK. The implications of this are two-fold. Firstly, users of these data for model evaluation should be mindful that emissions throughout the measurement period were not always at climatological levels. In addition to bulk concentration changes, pollution properties such as particulate size and composition may also have been different during these periods. While this does not negate the use of these data for some aspects of model evaluation, it certainly cautions against their use to assess quantitative performance of models driven using standard climatological emissions. Secondly, and more positively from a scientific perspective, as the database includes observations covering pre-, during and post-lockdown periods, it presents a unique and valuable resource with which to further explore changes in atmospheric composition over the UK associated with reduced emissions during the COVID-19-impacted period.

## 2 MOASA capability

The MOASA is a Cessna-421 aircraft based at Bournemouth airport, operated by Alto Aerospace Ltd for the Met Office (Fig 1). The MOASA is instrumented to allow airborne measurement of key air quality-relevant aerosol and gas phase pollutants; namely gaseous nitrogen dioxide ($NO_2$), ozone ($O_3$), sulphur dioxide ($SO_2$), and fine mode aerosol ($PM_{2.5}$, determined indirectly from measurements of the aerosol size distribution). The



fine mode aerosol is also characterised in terms of optical absorption and scattering properties. This section provides detailed description of the MOASA instruments and related quality assurance protocols.

**2.1 Instrumentation – general setup**

Instruments are situated in the cabin, the front hold of the aircraft and under the wings. Wing-mounted probes include an Aircraft-Integrated Meteorological Measurement System (AIMMS, Aventech) instrument that
provides real-time ambient meteorological data including temperature, humidity, pressure, three-dimensional winds (speed, direction, vertical) as well as latitude, longitude and (GPS) altitude. The aircraft also includes a wing-mounted Cloud, Aerosol and Precipitation Spectrometer with Particle-By-Particle (CAPS-PBP, Droplet Measurement Technology (DMT)) though it does not form part of the air quality measurement suite and therefore is not discussed further here. Nitrogen dioxide, ozone and sulphur dioxide instruments are rack
mounted in the cabin and sample at 0.85, 1.8 and 0.5 litres per minute, respectively. All instruments have a 1 Hz sampling resolution, except for the $O_3$ monitor which samples at 0.5 Hz. Ambient gaseous samples are drawn from a stainless-steel air sample pipe that takes air from outside of the fuselage boundary layer through an on-rack PTFE headed sample pump (KNF N834.3FTE). Also within the cabin is a backscatter aerosol lidar (Leosphere) which is used operationally though does not form part of the core air quality measurement suite.
The starboard side nose bay compartment contains a custom-built 'Air Quality Box' (AQ Box) and a nephelometer (Ecotech, Aurora 3000) (Fig 2). The sample to each of the instruments in the front hold is controlled with actuated valves and volume flow controllers inside the AQ Box (see Appendix A for AQ Box flow schematic).

The AQ Box contains a Portable Optical Particle Spectrometer (POPS, Handix) and a Tricolour Absorption
Photometer (TAP, Brechtel, model 2901) and has the capability to sub-select only $PM_{2.5}$ sample aerosol for analysis. The sample into the AQ Box is from a Brechtel Iso-Kinetic inlet which samples at 6.35 litres per minute and has >95% sampling efficiency for particle diameters from 0.1 to 6 μm (Brechtel Manufacturing Inc, 2011). The $PM_{2.5}$ sample flow is dried via two Perma Pure MD-700 driers, connected in series via a 180-degree bend. The sample then passes through an impactor with an aerodynamic cut point size of 2.5 μm, before being
split between the POPS (0.5 LPM (sample + sheath)), TAP (1 LPM) and the nephelometer (5 LPM) which is situated alongside the AQ Box. Measurements at the nephelometer and TAP inlet indicate the $PM_{2.5}$ sample relative humidity is typically below 20% and therefore the sample is a good representation of the dry $PM_{2.5}$ size distribution. Within the AQ Box the sample line temperature and pressure are also recorded.

Particle losses through the PM2.5 sampling lines have been estimated using open access particle loss calculation
software (Von Der Weiden et al., 2009) based on the tubing dimensions, flow characteristics and a representative particle density of 1.64 gcm$^{-3}$. This analysis has suggested losses downstream of the inlet of <17% for particle diameters in the range 0.1 - 3μm.

In addition to particle losses due to flow deposition, we have considered the extent to which loss of particle mass may occur due to evaporation of ammonium nitrate, $NH_4NO_3$, a semi-volatile aerosol component that
readily repartitions between condensed and gas phases upon changes in temperature and humidity (Nowak et al., 2010, Langridge et al., 2012, Morgan et al 2010). To determine the fractional loss of $NH_4NO_3$ during MOASA sampling, a kinetic model of the $NH_4NO_3$ evaporation process (based on the approach of Fuchs and Stutugin



(1971), as implemented by Dassios and Pandis, 1999) was used to calculate the rate of change in diameter of polydisperse $NH_4NO_3$ particles through the MOASA flow system. The model unsurprisingly showed that the

loss of particulate nitrate had a strong temperature dependence and varied dynamically as a function of time. Total mass losses during the MOASA sampling residence time of 2 seconds and at a representative sampling temperature of 30ºC were approximately 7%. The $NH_4NO_3$ losses showed a weak dependence on pressure and relative humidity, with absolute losses increasing by only 2% at 500mb compared to 100mb and by approximately 2% over the relative humidity (RH) range 10-50% (where in-flight $PM_{2.5}$ sample RH was

typically below 20%). Although evaporative loss of $NH_4NO_3$ during MOASA sampling will vary on a case-by-case basis, for representative conditions this work confirms that the loss is small and likely less than 7%.

The AQ box also allows for measurement of the aerosol population without particle size selection or drying, however this mode of operation has not been utilised in this work and is therefore not described further.

**2.2 Nitrogen dioxide**

A Cavity Attenuated Phase Shift Spectrometer Nitrogen Dioxide detector (Aerodyne Research Inc, referred to here as $NO_2$CAPS to avoid confusion with the Cloud and Aerosol Precipitation Spectrometer, CAPS) was repackaged in-house, from a 5U, 12 kg to a 3U, 9.7 kg 19" rack-mounted unit to optimize volume and weight for airborne use. The analyser monitors ambient atmospheric $NO_2$ concentrations up to 3000 ppbv (parts per billion by volume) using a 450 nm LED based absorption spectrometer utilizing cavity attenuated phase shift

spectroscopy (Kebabian et al., 2005). A comprehensive review of the theory of operation is detailed in Kebabian et al., 2005. The $NO_2$CAPS analyser has been shown to be insensitive to other nitro-containing species and variability in ambient aerosol, humidity and other trace atmospheric species (Kebabian et al., 2005, Aerodyne Research, n.d.).

While some cavity-based absorption techniques are often referred to as calibration free (Langridge et al., 2008),

this feature relies on knowledge of the variation in absorption cross-section across the spectral range of the light source being used. Given the broadband nature of the $NO_2$CAPS light source, which is difficult to characterise accurately and may be subject to change over time, we chose to undertake routine direct calibration of the instrument. As such, full multi-point calibrations are carried out annually at the National Centre for Atmospheric Science (NCAS) Atmospheric Measurement and Observation Facility (AMOF) COZI-lab at the University of

York. Here, a multi-gas calibrator is used to dilute a high concentration NO standard into zero air (grade Pure Air Generator (PAG) 001) at varying levels. Ozone is added in excess to ensure full conversion of NO to $NO_2$. Seven concentration levels are used, and zero checks are also carried out. Calibration coefficients are determined from linear fits and applied to the $NO_2$CAPS during data post-processing.

**2.2.1 NO₂ analyser baseline pressure dependency correction**

During normal operation, the $NO_2$CAPS analyser periodically establishes a baseline to account for the optical losses associated with light transmission by the cavity mirrors (which depend both on mirror cleanliness and alignment) and Rayleigh scattering of light by air (Kebabian et al., 2005). This is achieved by passing $NO_2$ free air through the analyser every 15 minutes (automated). The standard $NO_2$CAPS software then applies a constant baseline correction based on these periodic measurements for the sampling segment that follows. For variable-



pressure aircraft operation, this approach is not adequate as changes in Rayleigh scattering that accompany pressure changes lead to shifts in the instrument baseline between filter periods.

To account for these changes, a new correction scheme has been developed. During post processing, the pressure dependence of the baseline is determined by applying a linear fit to the pressure variation in Rayleigh-corrected filtered-air measurements recorded across the full flight. This dependence is used to calculate a new

time-varying baseline based on sample pressure measurements alone. This baseline is then used to recalculate the $NO_2$ concentration across the flight.  Spikes due to valve switches are also removed from the data series at this stage.

Figure 3 shows raw (red) and processed (blue) $NO_2$ concentration during flight M304 in November 2021, where

the $NO_2$CAPS sample inlet was fitted with a zero-air filter such that measurements were sensitive only to baseline changes. Following take-off at 11:52:00 the aircraft climbed to an altitude of 5.5 km resulting in an ambient pressure change of 509 mb and a $NO_2$CAPS measurement-cell pressure change of 250 mb. The profile shows corrected data is markedly more stable in comparison to the raw data and suggests a mean error in $NO_2$ concentration due to pressure-dependent baseline corrections of $\pm$ 0.09 ppbv (data averaged over 10s intervals).

The oscillations seen in the data during the filter test are an artefact of the filter, which impacted performance of the instrument pump. During a separate zero-air test experiment, the sensitivity of the $NO_2$CAPS was derived to be $0.17 \pm 0.14\sigma$ ppbv (data also averaged over 10s intervals). As such, following correction, $NO_2$CAPS pressure sensitivity is not considered a significant source of uncertainty for aircraft $NO_2$CAPS observations.

**2.3 Ozone**

A dual beam ozone monitor (2B Tech, model 205) enables measurements of atmospheric ozone up to 100 ppmv (parts per million by volume). Measurements are based on the absorption of ultraviolet (UV) light at 254 nm in two absorption cells, one with ozone-scrubbed (zero) air and one with un-scrubbed (sample) air from which the Beer Lambert law can be used to determine ozone concentration. Instrument sensitivity, empirically derived by sampling filtered air at 0.5 Hz during a test flight, is $2.9 \pm 0.4$ $\sigma$. The monitor is calibrated annually at the NCAS

AMOF COZI-lab where the instrument is compared with a NIST-traceable standard ozone spectrometer over a wide range of ozone mixing ratios. These results are used to calibrate the ozone monitor with respect to gain and sensitivity which are applied to the instrument directly.

A known but not widely recognized issue with UV absorption ozone monitors is that rapid changes in humidity (as may occur during airborne ascents and descents) can cause a large zero shift. This is due to modulation of

humidity of the sample stream by the ozone scrubber which can cause the humidity in the sampling and zero cells to go out of equilibrium. To equilibrate the humidity, Nafion tubes known as DewLines are used in the 2B Tech monitor (Dewline, n.d., Wilson and Birks, 2006). Biases may become apparent should the DewLines stop working effectively and thus, following some initial issues with negative calculated ozone values during MOASA measurements (impacting the first 7 flights), the Dewlines were regularly replaced.

**2.4 Sulphur dioxide**

A pulsed florescence $SO_2$ analyser (Thermo Scientific, 43i Trace Level-Enhanced) detects sulphur dioxide up to 1000 ppbv. It operates on the principle that $SO_2$ molecules fluoresce following absorption of ultraviolet (UV)



light, with the fluorescence intensity proportional to the number of $SO_2$ molecules in the air sample (Beecken et al., 2014). Instrument sensitivity was empirically determined using zero-air checks to be $0.90 \pm 0.26$ σ ppb

(averaged over 10s intervals). The $SO_2$ instrument is calibrated (zero and span) monthly in the field using an 863 ppb BOC Alpha Standard.

### 2.5 Aerosol scattering

A multi-wavelength integrating nephelometer (Ecotech, Aurora 3000) measures the light scattering coefficient of the aerosol population in both forward and back-scatter directions. It uses three high powered LED sources

operating at wavelengths of 450, 525 and 635 nm.

Instrument sensitivity, determined from baseline statistics when sampling filtered air over 30 minutes at wavelengths 450, 525, and 635 nm was $0.05\pm 0.51$σ, $0.10\pm0.55$ σ and $0.01\pm0.69$ σ $Mm^{-1}$ for total scattering, and $0.21\pm 0.95$ σ, $0.07\pm 0.49$ σ and $0.14\pm0.55$ σ $Mm^{-1}$ for backscattering, respectively (data averaged over 10 s intervals). This falls within the manufacturer specified sensitivity of <0.3 $Mm^{-1}$. A monthly $CO_2$ calibration and

annual in-house service are completed for the nephelometer as per manufacturer procedures (Ecotech, 2009).

Uncertainties in scattering measurements using the nephelometer are dependent on sample flow (empirically derived over all flights as < 0.05%), the uncertainty of calibration, inhomogeneities in Lambertian angular illumination, and truncation of light due to cell geometry. Corrections for angular truncation and non-Lambertian light source effects are applied according to the recommendations of Müller et al., 2011.

Müller et al., 2011 empirically calculated an uncertainty of 4% (450 nm), 2% (525 nm) and 5% (635 nm) for total scattering, and 7% (450 nm), 3% (525 nm) and 11% (635 nm) for total backscatter, which are adopted here. The signal to noise ratio for backscattering is worse compared to total scattering, since the backscattering signal is about one order of magnitude smaller than the total scattering signal for ambient air (Müller et al., 2011).

### 2.6 Aerosol absorption

Aerosol absorption is measured using a Tricolor Absorption Photometer (TAP, Brechtel, model 2901). The TAP is a 3-wavelength (467, 528, 652 nm) filter based absorption photometer which derives real-time aerosol light absorption from the difference in light transmission measured between two 47 mm diameter Pallflex (E70-2075W) glass-fibre filter spots, one of which receives particle laden air and the second of which receives aerosol-filtered air (Davies et al., 2019, Bond et al., 1999, Perim De Faria et al., 2021 and Ogren et al., 2017).

The TAP employs empirical corrections to account for scattering effects that complicate the derivation of aerosol absorption from filter transmission measurements. The theory of operation and characterisation of the TAP is given in Ogren et al., 2017, Davies et al., 2019 (where it is previously known as a `CLAP').

Mean 1σ detection limits of the MOASA TAP, empirically derived by sampling filtered air and averaging over 60 seconds, are 0.22, 0.18 and 0.26 $Mm^{-1}$ at wavelengths of 652, 528 and 467 nm, respectively. These values are

in line with the manufacturer provided noise level characterisation of 0.20 $Mm^{-1}$ over the same integration time.

The errors in absorption measurements from filter based photometry are dominated by uncertainties in the empirical scattering corrections, but also have contributions from uncertainties in the spectral response of the light source (±1-2 nm (Ogren et al., 2017)), sample flow rate (<1% (Ogren et al., 2017)), filter spot size and the

penetration depth of particles within the filter matrix (Bond et al., 1999, Davies et al., 2019, Müller et al., 2014, Virkkula, 2010, Ogren et al., 2017). Internal particle losses within the instrument flow system due to diffusion, impaction and sedimentation are estimated to be < 1% for particles with diameters in the range 0.03–2.5 µm (Davies et al., 2019, Ogren et al., 2017). To minimise the effects of instrument noise observed in-flight, a low-pass filter is applied to raw data with a cut-off frequency of 0.08 Hz although this had minimal impact on optical properties derived from these data.

We apply scattering corrections to the low-pass-corrected TAP data using the Virkkula, 2010 correction scheme which relies on simultaneous measurements of the light scattering coefficient, which in this case are provided by the nephelometer. The correction scheme is implemented as described by Davies et al., 2019. Ogren et al., 2017 provided an estimate of the accuracy of TAP absorption measurements of 30% and this value is adopted here. However, as summarised by Davies et al., 2019, given the empirical nature of filter-based correction schemes and strong source and wavelength dependencies, these correction schemes are unlikely to fully bound uncertainties associated with filter-based absorption measurements.

### 2.7 Aerosol size distributions

A portable optical particle counter (POPS, Handix) measures the size of dried particles predominantly in the accumulation mode (approximately 0.1 um < d < 1 um)(Haywood, 2008)using a light scattering technique. The POPS uses a spherical mirror to collect a fraction of light scattered sideways (38 – 142 degrees) by individual particles traversing a 405 nm laser beam. The scattered light is directed to a photomultiplier tube, the signal from which is digitised and placed into one of 32 bins that are spaced logarithmically in scattering amplitude space. For a given laser power, the measured scattering amplitude is determined by the particle size, shape, and index of refraction (IOR), thus allowing the bin boundaries to be converted to effective particle size subject to assumptions about shape and optical properties. In addition to particle size, given the POPS is a single particle instrument, it also provides a measure of the total particle number within its detection size range. A comprehensive review of POPS theory of operation is provided by Gao et al. (2016).

### 2.7.1 Calibration

Particle sizing by the POPS is calibrated by measuring the scattering amplitude of atomised NIST traceable polystyrene latex (PSL) spheres of known size, spherical shape and IOR (Rosenberg et al., 2012, Peers et al., 2019, Gao et al., 2016). Calibrations use 10 discrete sizes of PSL between 0.15 and 3 µm. The PSL are atomised and dried prior to entering the POPS sample inlet. PSL sizes between 0.15 and 0.70 µm are, where possible, also passed through a differential mobility analyser (DMA, TSI 3082 Electrostatic Classifier) in order to help minimise the impacts of contaminants from the PSL generation process.

For each PSL diameter, Mie theory is used to calculate the particle scattering cross section (Fig 4), using a PSL IOR at 405nm of 1.615+0.001j (Gao et al., 2016). Linear regression is then used to fit the relationship between the POPS-measured scattering amplitude and the theoretical PSL scattering amplitude (see Appendix B) (Rosenberg et al., 2012). The error in response is determined from the standard error in the mean for each 15 second period of sampling, averaged over the duration of the PSL run. The error in PSL diameter is the NIST-certified range of the PSL diameter. The linear regression function is used to assign calibrated scattering amplitudes to the designated POPS bin boundaries. At this point, the POPS measurements are calibrated.



To size ambient particles, it is necessary to convert the bin boundaries to equivalent diameters for particles with different optical properties. The impact of particle index of refraction on the POPS response is shown in Fig 4 which shows the relationship between particle diameter and theoretical POPS response for both PSL's and particles representative of urban sampling. To account for the significant differences seen, we again apply Mie theory. The calibrated POPS bin boundaries in scattering cross section space are converted to diameter space based on Mie calculations. These calculations integrate scattering over the angular range of collection angles of the POPS and use an estimate of the ambient particle IOR (further details below) (Rosenberg et al., 2012, Gao et al., 2016). To overcome inherent Mie resonance oscillations in calculated scattering signals (where Dp > 600 nm in Fig 4), which result in non-monotonic behaviour with increasing particle diameter (van de Hulst 1981, Gao et al., 2016, Rosenberg et al., 2012), each Mie response curve is smoothed using spline interpolation (Hagan and Kroll, 2020). As particle morphology and inter- and intra- particle homogeneity of the ambient sample are unknown, an assumption of spherical, homogeneous particles is implicit to the application of this Mie theory-based approach.

### 2.7.2 Index of Refraction

The IOR of the aerosol sample used for determination of POPS bins boundaries for ambient sampling is estimated using the method described in Liu and Daum, 2000 and Peers et al., 2019. This is an iterative approach whereby the single scattering albedo (the wavelength dependent ratio of aerosol scattering to total extinction, $\omega 0$) is calculated from the dry POPS particle size distribution ($\omega 0_{psd}$, $\lambda$ = 405 nm) using an initial guess IOR and then compared to the measured single scattering albedo at 405 nm derived from independent observations from the MOASA nephelometer and TAP ($\omega 0_{nt}$). The IOR is then adjusted iteratively until acceptable closure is reached between calculated and measured $\omega 0$, noting that the POPS bin boundaries are adjusted upon each iteration.

This process is summarised in Fig 5 and more detail, including a case study, is in Appendix 5.

A strength of the MOASA data set is that the POPS, TAP and nephelometer all share a common sample inlet, which reduces the potential source of sampling bias that may impact this analysis. Further, to minimise differences in sampling volumes and response times, all $\omega 0$ calculations are performed using 30 second averaged data and only data from straight and level runs (SLR, flight transects at approximate constant altitude and velocity) of at least 3 minutes duration are included. The iterative IOR analysis step is performed on the flight-mean of these SLR data. While this approach does not allow in-flight variability to be accounted for, it minimises potential for erroneous impacts on the POPS size distribution arising from noise and uncertainty in the $\omega 0$ measurements, which can be large at low aerosol loading levels. The flight-average approach adopted here has been shown to lead to modest errors in particle diameter of <10% compared to analysis at finer temporal scales (see Appendix C, case study). We also note while the IOR derived here provides closure between MOASA optical and size distribution instruments, it is subject to potential uncertainties that caution against its use as an accurate measure of the true ambient particle IOR (Frie and Bahreini, 2021).

### 2.7.3 Size distribution uncertainties

A review of uncertainties for the POPS instrument is given in Gao et al. (2016). For particle number measurements, the main source of uncertainty for particles within the instrument's size detection range is the



sample flow rate. Gao et al. (2016) report a nominal sample flow rate of 3 cm$^3$ s$^{-1}$ with an upper limit of 6.67 cm$^3$ s$^{-1}$ and associated error of <10 % (personal communication, Handix, October 2020). For the MOASA POPS the sample flow over all flights ranged from 2.7 to 5.9 cm$^3$ s$^{-1}$ (data averaged over 10s intervals). The higher values arose due to flow system cross-interference issues that generated flow noise impacting the first 11 MOASA flights, following which the source of noise was removed and a more representative range of normal

operation is 2.9 cm$^3$ s$^{-1}$ ± 3.2%.

Coincidence errors, whereby two or more particles traverse the laser beam at the same time leading to sizing errors, are a common feature of all optical particle counters when used in high aerosol loading environments. The impact of coincidence errors on the MOASA POPS observations are addressed during data processing by flagging all data where particle concentrations exceed 7000 cm$^3$/s   (McMeeking, 2020, personal

communication).

Particle sizing uncertainties arise from a number of sources, including scattering amplitude measurement uncertainty (leading to an estimated 3% 1σ sizing error for 500 nm particles) and laser intensity instability (±3 % diameter sizing error for temperatures from 43 to 46 °C). In addition, for reasons already discussed above, uncertainty in the IOR of particles being measured also impact uncertainty in particle sizing. Gao et al. (2016)

used a theoretical ambient aerosol population to investigate the potential magnitude of this error. They assessed the accuracy in the location and width of lognormal fits to both a theoretical population fine mode (10% and 10% respectively) and coarse mode (1.4% and 19% respectively). These uncertainties were propagated to derive an estimated uncertainty in the total particle volume of 19%.  Though based on a single theoretical ambient size distribution, this analysis provides an indication of the magnitude of error arising from IOR variation. For

MOASA POPS-derived size distributions, it is likely to provide an upper indication of the error, given that efforts to correct the POPS bin boundaries based on the iterative IOR method described above should serve to improve sizing accuracy.

Based on the information above, an upper estimate for the error in total particle volume from POPS measurements (required for subsequent calculation of particle mass) is derived by combining in quadrature

contributions from IOR/scattering (19%), sample flow (3.2%) and laser amplitude (6%) to yield an uncertainty of 20%.

**2.8 Determination of mass concentration (PM2.5)**

To calculate particulate mass, we convert the calibrated, IOR-corrected POPS particle size distributions to volume distributions, and subsequently mass distributions by assuming a fixed particle density. The total mass is

then calculated by integrating across the distribution within the PM2.5 size range. Calculations are performed on 10 second averaged data and work on the basis of fitting lognormal functions to the measured distributions to represent a fine and coarse mode (the dashed line in Fig 6 shows the combined lognormal modes from a straight and level run during flight M270 on 15th September 2020). This approach serves to reduce the impact of residual structure from Mie resonances in the POPS distribution on mass derivations.

The selection of an appropriate particle density for converting volume to mass is an important part of the above analysis. The composition and therefore density of ambient aerosol varies dynamically in the atmosphere (Wang et al., 2009, Crilley et al., 2020). In the absence of co-located aerosol composition observations on MOASA, we


apply a fixed density to all data of 1.64 ± 0.07 (1σ) gcm$^3$. This value is derived by weight-averaging the densities of PM$_{2.5}$ aerosol components measured during a range of UK field experiments, as detailed in
Appendix D.

The total uncertainty in the determined PM$_{2.5}$ mass concentration, estimated by combining uncertainties in the measured particle volume (20%) and the assumed particle density (4.2%), is 20.4% and thus dominated by the volume error.

### 3 Flight Planning

The MOASA air quality flight strategy was based on flying a series of repeated sorties, each designed to provide data suitable for different aspects of model evaluation work. On a week-to-week basis, sorties were selected based on the prevailing weather conditions and any required modifications to flight plans are made at that time. This section describes the rationale behind each of the sortie types, together with a summary of flight activities.

Given the MOASA home base is at Bournemouth on the south coast of the UK, operations have predominantly
focused on sampling over the south of the UK. This includes work over the English Channel (e.g., sampling transboundary pollution), over varied land-use types (urban and rural) including pollution hotspots such as London, and over isolated source regions such as docks and industrial sites. In addition to regular sorties, in June and July 2021 the MOASA also participated in an Intensive Observation Period (IOP) in conjunction with ground based air quality super-sites located in London, Birmingham and Manchester (UKRI, 2021, OSCA,
2020). All flights are performed within operational airspace regulations which limit minimum and maximum flight levels. Observations are mostly in the boundary layer and, as shown in Fig 8, typically near or below 1 km GPS altitude. The lowest altitudes (0.15 km minimum) are permitted in offshore and rural areas, whereas minimum altitudes in urban areas (or in regions with significant topography or obstacles like masts or chimneys) are limited to > 0.3 km. Where possible profile measurements extending into the free troposphere are also
collected, which allow the boundary layer height to be determined in addition to sampling of aged and/or transported pollutants.

In terms of meteorology, conditions representative of both the general background environment and elevated pollution events have been targeted. As the southern UK has a maritime climate, with the frequent passage of mobile low-pressure systems from the North Atlantic, conditions in the operating area are not always conducive
to the build-up of pollution. For the targeting of elevated pollution conditions, synoptic high-pressure conditions with light winds and little cloud/precipitation are favoured. Strong sunshine and elevated temperatures are also conducive to the production and build-up of pollutants such as ozone and as such, high pollution events tend to be more frequent and severe in the summer (Savage et al., 2013).

### 3.1 Ground Network Survey

Ground Network Survey sorties describe two flight patterns that sample both rural and urban background regional pollution at various altitudes. One flight pattern is focused on the southwestern UK (Fig 7, panel A1) and the other on the eastern UK (Fig 7.A2). A particular feature of these sorties is that they overfly a number of AURN ground sites allowing pollutant concentrations at the surface to be compared to those aloft. Characterisation of pollution at regional scales is important for air quality model evaluation, particularly for


models operating at coarse resolutions such as AQUM, which encompass point-source emissions data but cannot accurately represent them in terms of location and concentration.

### 3.2 High-Density Plume Mapping

High Density Plume Mapping flights (Fig 7.B) use intensive model grid-box scale sampling to allow for assessment of the (often sub-grid in models) scale of pollutant variability in a high pollution region. Repeated
runs upwind, downwind and within the plume are performed at a range of altitudes. This sortie has primarily been flown over Port Talbot in South Wales, a heavily industrialised area and AQUM pollution hotspot, but has also been flown once north of Cambridge (east UK). In that case, horizontal transects sampling the plume at multiple altitudes downwind of the city were conducted.

### 3.3 South Coast Survey

South Coast Surveys were flown onshore and offshore along the south coast of the UK, typically from Dartmoor National Park in the western UK to Eastbourne in the east (Fig 7.C). These surveys have been flown under background and polluted southerly flows to characterise transboundary and long-range transport of pollutants from continental Europe. In late 2019, a persistent emissions hot spot (primarily $PM_{2.5}$ and $SO_2$) was seen in the AQUM forecasts, potentially originating from ships in Southampton Docks. Therefore, from late 2019 onwards,
overflights of the Solent and Southampton Waters were added to the stock sortie.

### 3.4 Coastal Transition Survey

The coastal transition sortie (Fig 7.D) also operates along the south coast of the UK. The primary distinction from the south coast survey was a zigzag manoeuvre whereby observations across the land-to-sea transition are repeatedly sampled. The objective for this sortie is to obtain data for benchmarking model performance across
the land-sea interface where strong gradients in humidity and temperature can impact forecast pollution fields. In later flights, these surveys have also been extended eastwards to encompass the Dover Straights to allow sampling of pollutants transported from industrial activities around the Dunkirk region of northern France, which is another emissions hotspot that can lead to strong pollutant transport over the UK when meteorological conditions permit.

### 3.5 London City Survey

Circumnavigational flights of London (Fig 7.E) were performed during high and low pollutant loadings to characterise city scale emission and dispersion of pollutants from the heavily populated, commercial, and industrial Greater London area. Busy air space and air traffic control due to the close proximity to major airports (Gatwick, London City, Heathrow) restrict the operational area of the MOASA. Broadly, following a short
transit to Reading, the sortie takes the MOASA clockwise following the M25 London orbital motorway, which encircles Greater London. Missed approaches are frequently performed at Elstree airfield to the north and Biggin Hill airfield to the southeast.

A substantial decrease in air traffic during the COVID-19 pandemic provided a unique opportunity to fly at low level (approx. 1000 ft) over central London. This central city sampling was added to the stock sortie in
November 2020 and became the primary sortie for flights during the COVID-19 pandemic. The central London overpass follows the Thames River to approximately 0.087°W where it deviates south-westerly to comply with



air traffic control restrictions. During later flights, north-south and/or east-west transects were also completed to observe the urban heat island effect on boundary layer height.  During the IOP's in June-July 2021, and January-February 2022 MOASA observations were also made close to the surface air-quality IOP supersite (stars, Fig 7.E).

### 3.6 Birmingham and Manchester IOP

During Clean Air ground based IOP's in June-July 2021, and January-March 2022 MOASA observations were also made over Birmingham (Fig 7.F) and Manchester (Fig 7.G).  These city scale sorties were tailored to best suit meteorological conditions on the flight day, and typically involved circumnavigational orbits, or box patterns over the cities at altitudes ranging from approximately 0.3 to 0.9 km and/or runs north to south, up wind and downwind of the city and supersite. Passes directly overhead of the Birmingham and Manchester ground supersites (stars, Fig 7f and 7g) were made at each altitude, when possible. During the IOP, MOASA operated both in the morning and late afternoon, allowing observation of the build-up of regional scale pollutants over the day. Further MOASA flights in these regions are anticipated during a second ground based IOP planned for winter 2021/22.

### 3.7 Summary

63 flight sorties were flown between June 2019 to April 2022, comprising over 150 hours of atmospheric sampling. Flight details are summarised in table 1. Figure 8 shows horizontal and vertical spatial coverage of flights over the Clean Air campaign.

### 3.8 The MOASA measurement database

Datasets obtained during the MOASA Clean Air project are openly available from the Centre for Environmental Data Archive (CEDA) "Collection of airborne atmospheric measurements for the MOASA Clean Air project" repository (DOI: 10.5285/0aa1ec0cf18e4065bdae8ae39260fe7d).

Data files are NetCDF format and contain observations from the NO$_2$CAPS (NO$_2$, ppbv, 1Hz), Ozone monitor (O$_3$, 0.5 Hz, ppbv), SO$_2$ analyser (SO$_2$, ppbv, 1Hz), nephelometer (light scattering, Mm$^{-1}$, 1 Hz), TAP (light absorption, Mm$^{-1}$, 1Hz), POPS (particle counts, and calibrated, IOR corrected particle concentration, total mass (µg m$^{-3}$ / bin) and PM$_{2.5}$ (µg m$^{-3}$), 1 Hz), as well as meteorological parameters observed by the AIMMS-20 (ambient temperature (ºC), relative humidity (%), pressure (hPa) and wind speed (m/s) and wind direction (degree), 1 Hz). Each instrument parameter is presented as a time synchronised, three-dimensionally geo-located time-series, with calibrations and corrections applied (where applicable). Each instrument parameter has a standard name, long name, unit and measurement frequency (compliant with Climate and Forecast (CF) naming conventions where possible), which are listed in Appendix 7. Some, but not all, also have a comment, minimum and maximum limits and/or a positive attribute. Each variable has the coordinates of time, latitude, longitude and altitude. Measurements from all instruments are reported at ambient pressure and temperature.

To ensure optimal traceability and transparency of data, comprehensive metadata is included in the NetCDF which details any calibration constants and/or corrections applied to data alongside general information about the data, such as contacts, acronyms and references. Data is range checked to ensure observations fall inside the recommended operational limits of the instrument and outliers to these limits are flagged. The standard flag



format is the parameter name, post fixed with '_FLAG'. The three flag values are: 0 = good_data, 1 =
outside_valid_ranges, and 2 = sensor_nonfunctional. For flag=1, the valid ranges are given in the variable
metadata. Each flag parameter has standard name, frequency, flag value and flag meaning attributes. Derived
variables (for example, $PM_{2.5}$ or Angstrom exponents) do not have flags. The configuration file used to process
each flight data is available alongside the NetCDF as a text file and provides the range check limits and the
source of these limits. Records of all work done on the instruments (calibrations, cleaning, and maintenance) are
digitally recorded and available on request.

## 4 Flight data examples

This section provides a limited number of case studies applying the MOASA dataset to different scientific
applications. These examples are intended to showcase different uses of the database and are not intended as
comprehensive analyses in their own right. We present: i) a statistical analysis of the scales of pollutant
variability observed across the MOASA air quality dataset, ii) example use of the dataset for evaluation of a
regional air quality modelling system (AQUM), and iii) the vertical structure of pollutants observed during
repeated flight patterns over Greater London, including during the COVID-19 impacted period.

### 4.1 The spatial scales of pollutant variability

The evaluation of limited-resolution regional air quality models (such as AQUM with a 12km grid length) using
high resolution in-situ surface or airborne data, is complicated by the differences in spatial scale between the
two. While instrumentation may be capable of measurements at high precision and accuracy, these uncertainty
metrics may, or may not, provide criteria suitable for determining the degree to which models and observations
should agree. In many cases the magnitude of natural pollutant variability at scales that are sub-grid for models
provides an important additional consideration. With this in mind, in this section we use the MOASA Clean Air
database to assess how observed pollutant variability changes, on average, as a function of length scale, and how
this variability compares to fundamental instrument measurement precision.

We take a statistical approach that uses data from all MOASA SLRs, over 44 flights between July 2019 and July
2021. The number of SLRs per flight varies depending on the type of sortie flown, with a minimum of 2 and a
maximum of 11 (see table 1). The minimum permissible SLR length was capped at 3 minutes to ensure adequate
counting statistics. In total this yielded 240 SLRs representing 1,389 minutes of sampling and we focus here on
measurements of relative humidity, $NO_2$, $SO_2$ and total particle number concentration.

High temporal resolution datasets corresponding to each straight and level run (e.g., $SO_2$ in Fig 9), formed the
basis for the analysis. Measured values in each dataset were split into groups of equal size, with sizes
corresponding to equivalent ground distances ($d_{int}$) ranging from 0.42 km to 17 km, in 0.085 km (1 second)
intervals. The variability observed at each of these length scales was calculated by first calculating the standard
deviation ($\sigma$) of points within each group of data, before calculating the mean deviation across all groups in the
transect. In order to provide a more statistically robust indication of ambient variability than possible from a
single transect, the mean transect $\sigma$ for each $d_{int}$ was averaged across all flight transects to give the flight mean
variability (e.g., Fig 10 for flight M284). Further, the analysis was extended to all flights in the MOASA



database, with results presented in Fig 11 as probability density functions of the mean transect $\sigma$ at selected $d_{int}$ of 0.42, 0.85, 2.55, 5.10, 12.07, and 15.04 km.

Figure 10 shows that the variability in observed RH, $SO_2$, $NO_2$ and particulate counts increased as a function of sampling scale. This result is unsurprising given that natural variability can only increase when observing over greater spatial scales. Interestingly the increase is non-linear, showing rapid change over scales of 0.5-2 km

before levelling at scales towards 15 km. For reference, the AQUM grid length of 12km is marked on the plots (vertical dotted line). The horizontal dashed red lines on Fig 10 show the precision of measurements derived from ground-based zero tests (where available) where $SO_2 = 0.90 \pm 0.26 \sigma$ ppbv, particle counts = $2.95 \pm 0.74 \sigma$ counts and $NO_2 = 0.17$ ppbv $\pm 0.14$. It is clear that even at the smallest spatial scale of 0.42 km, instrument precision did not limit ability to sample the natural pollutant variability for this flight.

Figure 11 extends this analysis to the full MOASA database, providing probability density functions that indicate the range of variability observed at a number of fixed sampling scales. Of particular note, it is clear that measured variability in $SO_2$ was generally close to or below the noise limit of the MOASA instrumentation. Hence aside from cases of elevated emissions (such as flight M284, fig 10), instrument performance dominates observed $SO_2$ variability in the MOASA database. For RH, $NO_2$ and particle counts, the natural variability is

generally well sampled by the MOASA instrumentation. It is interesting to note how the peak position and width of the distributions changes upon moving to progressively longer sampling scales. These changes tell us how, in an average sense, we might expect model sub-grid variability to change as a function of grid box length. Changes are particularly marked for relative humidity and somewhat less so for $NO_2$ and particulate counts. Focussing on the 12km length scale relevant to AQUM, the upper ends of the distributions bound the (average)

sub-grid variability that we might expect model output to represent. For $NO_2$ this absolute variability is below 7.35 ppbv and for particulate counts below 2412.830 counts/second for 90% of data points.

### 4.3 Preliminary model evaluation

In this section we show examples from two flights illustrating how the MOASA Clean Air database can be used for model evaluation purposes. These flights are: M270 high density plume mapping on 15th September 2020,

selected to measure the vertical distribution of pollutants in the lower atmosphere north of Cambridge (52.2053° N, 0.1218° E) and M296, a Birmingham city survey as part of the IOP on 1st July 2021. Meteorological conditions for the flights are summarised in Fig 12. For M270, there were largely clear skies with light winds (<10 m/s) in the south east UK where sampling was undertaken, and high temperatures (The National Meteorological Library, 2020), conducive to the accumulation of pollutants in the boundary layer. M296 was

influenced by high pressure, light winds and thin broken cloud.

Case studies of the flight days have been run using the AQUM UK domain model. This is the same model configuration used for the operational air quality forecasting, but for these case studies, no statistical post-processing (Neal et al., 2014) has been applied to the model data. Each simulation has been run with a 7 day spin up period. No adjustments have been made to the emissions used by the model to account for changes in

activities during the COVID-19 restrictions. Model data points have been linearly interpolated using the time, latitude, longitude and altitude coordinates of the aircraft at 1 second frequency. The model and aircraft data along the flight tracks have then been averaged into 10 second, non-overlapping intervals.



**Ozone**

Large ozone biases are seen for both flights (Fig 13). The model data show large overprediction when compared
against the aircraft data at corresponding locations (mean model bias of 18.49 ppb and 48.93 ppb for M270 and
M296, respectively). It is of note that this model bias is expected to have been larger if the AQUM data was
produced using emissions modified for the COVID-19 pandemic (Grange et al., 2021). The bias appears to be
relatively consistent across the latitude and longitude ranges of the flights and does not show any particular
correlation with location. For M270, the bias is lowest near to the surface and increases with altitude up to
approximately 700 - 800 m, above which the bias decreases. This can be attributed to differences in modelled
and observed boundary layer height, which is discussed further in the following section.

Savage et al. (2013) also reported biases during a ground-site AQUM comparison. A statistical post-processing
routine using ground based observations is applied to the forecast model data in order to generate the operational
forecast and this is known to significantly improve predictions (Neal et al., 2014). It may be possible to use the
aircraft observations to help identify sources of model bias, in a similar process to the above, or to determine an
ozone bias correction factor that can be applied to the model data.

**Nitrogen Dioxide and boundary layer height**

Figure 14 shows comparison between the model and aircraft $NO_2$ data for vertically stacked transects conducted
during M270. The agreement is generally good (within ± 2 ppbv) below 650 m altitude, but the model shows
large under-prediction above this altitude. Temperature and relative humidity profiles measured by the aircraft
(not shown) indicate a boundary layer height of approximately 1100 m on this day, which corresponds with a
decrease in observed $NO_2$ concentration above this height. However, the average boundary layer height in the
model for the observed area is approximately 620 m. This significant under-prediction in boundary layer height
is responsible for the poor predication of $NO_2$ at elevated altitudes and elucidates the altitude dependence on the
M270 ozone model bias discussed in the previous section. This comparison indicates the value of evaluating
model performance throughout the atmospheric column and suggests that the good agreement of $NO_2$ seen at the
surface may in this case have been somewhat fortuitous.

**Nitrogen Dioxide concentrations around Birmingham**

Figure 15 shows model and observed $NO_2$ concentration throughout the first and fourth stacked box patterns
performed around Birmingham during M296. Strong variation is observed in $NO_2$ concentration aloft of the city,
including enhanced $NO_2$ at all altitudes (maximum 55.70, 49.44, 56.31 and 54.06 μg/m$^3$ $NO_2$ for circuits 1-4,
respectively. Circuits 2 and 3 not shown). The enhanced $NO_2$ plume is seen above the southwest quadrant of the
city during the lowest altitude circuit (circuit 1, 423 m, 11:23 to 11:43 UTC) and moves southeast with
increasing altitude, until the plume is observed primarily over the southeast quadrant of the city during the
highest altitude circuit (circuit 4, 657 m, 12:33 to 12:52). Light north-westerly winds (0 < 5 knots) associated
with the high-pressure system are observed in all circuits. The observed peak in $NO_2$ seems to be located
downwind of important sources (motorways and a heavily urbanised area). Comparison with surface-level
observations (boxes) show that the plume aloft has greater concentrations of $NO_2$. The AQUM model shows
little variation and comparatively low $NO_2$ concentration in all circuits above the city (maximum 14.44, 13.91,
11.43 and 10.33 ug/m$^3$ $NO_2$ for circuits 1-4, respectively), and a negative $NO_2$ model bias is evident at the



observed plume location (maximum difference of -44.26, -44.30, -49.22 and -49.79 ug/m³ $NO_2$ for circuits 1-4, respectively). This model bias is expected to have been larger if the AQUM data was produced using emissions modified for the COVID-19 pandemic (Grange et al., 2021). In consonance with the observations, the model also shows light north-westerly winds at all altitudes. Modelled $NO_2$ concentration is comparable to surface level $NO_2$ at the lowest altitude circuit and decreases imperceptibly with altitude.

Given the aircraft flight track is mostly within just four model grid boxes, variation in $NO_2$ concentration, point source emissions, influence from local meteorology or dispersion of pollutants due to local topography is not expected to be represented in fine detail in the model. The lack of any enhanced $NO_2$ at all levels of the model could be attributed to a multitude of reasons, such as $NO_2$ emissions being too low at the observed plume location or (given $NO_2$ aloft is observed to be higher than surface level during this flight) inaccurate representation of the vertical structure of the atmosphere, where layers aloft may have a build-up of pollutants due to the slack winds.

**4.4 Long term observations over London**

In this section we look at long term surface level and airborne $NO_2$ and $O_3$ data to illustrate how the two datasets can be combined to help characterise persistent trends in the temporal and vertical distribution of pollutants.

Figure 16 (a) and 16 (b) show long-term surface-level background $NO_2$ and $O_3$ observations (time series) alongside concurrent airborne $NO_2$ and $O_3$ observations (box plots) within the Greater London area. The median London AURN observations are the hourly median value across all (active for species) AURN sites within Greater London (an average of 6.6 $NO_2$ and 4.6 $O_3$ sampling sites for the duration shown). This median is then resampled on to a monthly timestep which assists in visualising long-term trends in the surface-based observations. A list of the AURN sites and site types included in the analysis is given in appendix E. Airborne observations from within the Greater London area are used from 21 and 22 flights for $NO_2$ and $O_3$, respectively, the flight numbers of which are given as the London sortie type in table 1. Median aircraft altitudes (figure 16.c) over the duration of the analysis were approximately 480 m and the difference in altitude between observations at the surface and aloft is within 860 m.

Higher concentrations of $O_3$ are observed aloft by the aircraft, where, further away from the surface sources of nitrogen oxides (NOx=NO+NO₂), $O_3$ can reform through the oxidation of NO to $NO_2$ with peroxy radicals and subsequent photolysis of $NO_2$ to form $O_3$ (Lee et al., 2020). As such, the increase in $O_3$ is coincident with a reduction of the observed $NO_2$ aloft, which, in addition to being reduced by chemical reaction, is also further away from sources (fossil fuel burning, traffic (Jones et al., 2021, Lee et al., 2020)). Here, the impact of external factors (meteorology, boundary layer height, seasonal changes, complex chemistry) are not discussed and is beyond the scope of this paper. However, the persistent difference between the surface-based observations and airborne observations aloft demonstrates the importance in quantifying the vertical structure of pollutants, so their transport to/from the surface and the associated complex chemistry can be better evaluated in models, potentially reducing the dependence on surface-level model bias corrections, such as those discussed in section 4.3.



The surface based measurements show that following lockdown on 26th March 2020, which saw a 75% reduction in road traffic across the UK (Lee et al., 2020), the concentration of surface-level $NO_2$ decreased at
all AURN sites within Greater London. Here, we calculate the mean hourly concentration for each site for the pre- and post- lockdown periods. We define the pre-lockdown period as 26th March 2018 to 25th March 2020 and the post-lockdown period as 26th March 2020 to 25th March 2022, where comparing like-for-like months pre- and post- lockdown minimises the impact of seasonality on the comparison. Individual site data for the two periods are shown in appendix E. Averaged across all sites in Greater London, the mean hourly $NO_2$
concentration decreased by 6.89 ug/m³ (24.74 %) following lockdown.

This can be contrasted with an average increase of 3.35 ug/m³ (8.12 %) in the hourly surface-level $O_3$ across all sites in Greater London following lockdown (calculated as per above, with individual site data shown in appendix E). The mean increase in $O_3$ is consistent with the reduction in NO emissions following lockdown acting to decrease the extent of chemical loss of $O_3$ through reaction with NO (Air Quality Expert Group, 2020).
However, the increase in surface level $O_3$ is not observed at all sites; of the 6 sites analysed, an increase is seen in the 5 urban background sites and a decrease is seen in the single suburban site. This suggests more complex changes in the production/distribution of $O_3$ in Greater London during the pandemic, consistent with literature on UK-wide surface-level $O_3$ during the pandemic (Jephcote et al., 2020, Lee et al., 2020, Air Quality Expert Group, 2020, Wyche et al., 2021) and further work is recommended on the effect of observing site location on
ozone production.

Throughout the pandemic, the start and end of lockdown periods were not clearly defined (restrictions were incrementally decreased in different locations on different timescales). As such it presents a complex timeline and, given individual flight observations were made over discrete time periods, perturbations in long-term trends in airborne $NO_2$ and $O_3$ due to COVID impacted emissions are not immediately evident. However, the
availability of airborne observations concurrent to this complex timeline presents a unique opportunity to examine, in depth, case studies of the three-dimensional distribution of emissions below climatological levels during the COVID-19 pandemic, as well as the subsequent recovery to 'normal' (pre-pandemic) emissions, which is beyond the scope of this paper.

**5 Conclusions and future plans**

A long-term, quality assured, dataset on the three-dimensional distribution of $NO_2$, $O_3$, $SO_2$, and fine mode $PM_{2.5}$ aerosol, including optical absorption and scattering properties, has been collected over the UK using the instrumented Met Office Atmospheric Survey Aircraft from 2019 to April 2022. Observations allow for the evaluation of regional air quality models such as AQUM. A description of the MOASA measurement platform and instrumentation is presented, along with details of flight plans, designed to allow repeatable, comparable
observation of pollutants.

63 flight sorties, totalling over 150 hours of sampling, were flown during the campaign. These flights include observations of city scale pollution over Birmingham and Manchester during two periods of intensive



observations in June-July 2021 and January-February 2022, as well as long-term (2019 to 2022) observations over London, including central London overpasses (from October 2020).

Analysis of relative humidity, total particle counts, $NO_2$ and $SO_2$ over the campaign shows that instrument precision did not limit the ability to sample the natural pollutant variability, with the exception of $SO_2$, where limited instrument sensitivity dominated in all but a few cases where enhanced $SO_2$ concentrations were encountered.

Preliminary comparison of aircraft observations and AQUM data show the utility of the MOASA Clean Air
database for air quality model evaluation work. For the two flights analysed (M270 and M296), we show several cases of model-observation discrepancy that provide handles for further investigation associated with biases in modelled $O_3$ and $NO_2$ concentrations and boundary layer height. We anticipate that in addition to evaluation work, the airborne dataset may also be useful for derivation of bias-correction factors that can be applied to model data during post processing.

Analysis of long-term airborne observations over Greater London from September 2019 to March 2022 show persistent differences in the vertical distribution of the pollutants that have not routinely been available to evaluate and develop air quality models before. Specifically, we show lower concentrations of $NO_2$ and higher concentrations of $O_3$ aloft. The database also presents a unique and valuable resource with which to explore changes in atmospheric composition associated with reduced emissions during the COVID-19-impacted period.
Analysis of long-term surface-level trends in the Greater London region show a decrease in $NO_2$ and an increase in $O_3$ following the mandated COVID-19 restrictions. The availability of concurrent airborne observations presents a unique opportunity to examine the three-dimensional distribution of the reduced emissions in detail during this time.

This paper serves as a reference for all future database users. The MOASA Clean Air database is comprised of
quality assured observations, presented in NetCDF format and is accompanied by robust metadata to ensure traceability and transparency of data. A Clean Air Data Framework is currently under development which will host the data. Whilst the framework is under development, data is available by request.

## 6 Author contribution

JK, AW, DT and JB instrumented the MOASA. JK, AW, KW, JL, NN, ES and AM developed and planned
flight sorties, and JK, AW and KW carried them out. AM designed, developed, and applied the post-flight quality assurance and processing software, with nephelometer, TAP and $NO_2$ (including development of the pressure dependent baseline correction) modules adapted from original code by Kate Szpek, Nick Davies/JL, and JL respectively. ES provided the AQUM model data and ES and BD assisted the observation/model comparison. AM, JL and SA conceptualised the analysis (Sect 4) and AM performed the formal analysis. MH
devised and wrote the SPF Clean Air research programme of which the MOASA flights are an integral part, acquired the financial support for the project and also contributed to the original concept of a prolonged observation campaign. NN lead the conceptualization of MOASA involvement in the IOP's. AM prepared the manuscript with contributions from JL, ES, JK, KW, SA and MH.



The authors declare that they have no conflict of interest.

**7 Acknowledgments**

The MOASA Clean Air project is supported by the Clean Air programme which is jointly delivered by the Natural Environment Research Council (NERC) and the Met Office, with the Economic and Social Research Council (ESRC), Engineering and Physical Sciences Research Council (EPSRC), Innovate UK, Medical Research Council (MRC), National Physical Laboratory (NPL), Science and Technology Facilities Research

Council (STFC), Department for Environment, Food and Rural Affairs (Defra), Department for Health and Social Care (DHSC), Department for Transport (DfT), Scottish Government and Welsh Government.

The authors acknowledge and thank Alto Aerospace for efforts in delivering flight operations, the Met Office Guidance Unit who have supported flight planning throughout, Kate Szpek and Nick Davies, who's nephelometer and TAP (respectively) processing software were adapted for this work, and Debbie O'Sullivan

for creating Fig 1.

All map tiles by Stamen Design (https://stamen.com/), under CC BY 3.0 (https://creativecommons.org/licenses/by/3.0/). Data by OpenStreetMap (https://openstreetmap.org), under the Open Database License (https://www.openstreetmap.org/copyright).

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

**Appendix A: AQ Box schematic**

**Appendix B: POPS calibration**

**Appendix C: Index of refraction corrections**

$\omega 0_{nt}$ is determined by calculating the average single scattering albedo over the same flight transect as $\omega 0_{psd}$. First, the Virkkula-corrected TAP (absorption) data is smoothed to a 10 second triangular window to match the Muller-corrected nephelometer (scattering) data. The scattering and absorption Ångström exponents (SAE and AAE, respectively), calculated as per equation C1, were used to adjust the multi-wavelength nephelometer ($\lambda$ = 635, 525 and 450 nm) and TAP ($\lambda$ = 652, 528 and 467 nm) instruments to the POPS wavelength ($\lambda$ = 405 nm)
using equation C2 (Perim De Faria et al., 2021). Uncertainties in derivation of AAE (from potential asynchronous sampling response times and flow rates) were reduced by applying maximum and minimum bounds estimated by considering the extremes of expected ambient AE values. Here, the AAE upper and lower bounds are 3 and 0.7, respectively, AAE is removed when raw red absorption < 1 Mm$^{-1}$ and the AAE is set to 1.5 if the difference between absorption channels is < 1 Mm$^{-1}$. For the SAE, upper and lower bounds are 2.5 and
0.5, respectively, SAE is removed when raw red absorption < 10 Mm$^{-1}$ and the AAE is set to 0.5 if the difference between scattering channels is < 1 Mm$^{-1}$. The data is then further averaged over 30 seconds to minimise variability from instrument noise/precision and any mismatch of data. To minimise uncertainties in wavelength correction using the Ångström exponents, $\omega 0_{nt}$ is derived from the blue wavelengths only, using equation C3.

$$AE = \frac{-\log\left(\frac{AOCt_{\lambda 1}}{AOC_{\lambda 2}}\right)}{\log\left(\frac{\lambda 1}{\lambda 2}\right)}$$

Equation C1: where AE is the Ångström exponent, AOC = Aerosol Optical coefficient (scattering or absorption) and $\lambda_1$ and $\lambda_2$ are wavelengths pairs.

$$AOC_{\lambda_{405}} = AOC_{\lambda i}\left(\frac{\lambda_{405}}{\lambda_i}\right)^{-A}$$

Equation C2: where $\lambda_{405}$ is the POPS wavelength (nm), $\lambda_i$ is the wavelength of the given scattering or
absorption coefficient and AE is the Ångström exponent.





$$\omega 0_{nt} = \frac{\overline{scat\_blue}_{\lambda_{405}}}{\overline{scat\_blue}_{\lambda_{405}} + \overline{abs\_blue}_{\lambda_{405}}}$$

Equation C3: where the bar indicates the 30 second rolling average, for scattering (scat) and absorption (abs) for the blue wavelength nephelometer and TAP channels, converted to POPS wavelength ($\lambda_{405}$).

Determining $\omega 0$ using separate instruments with different uncertainties and principles can lead to potentially significant errors and biases (Perim De Faria et al., 2021). The uncertainty in the $\omega 0_{nt}$ calculations is related to the corresponding uncertainties in the scattering and absorption coefficients (Peers et al., 2019) measured by the nephelometer (4% at 450 nm, 2% at 525 nm and 5% at 635 nm, Müller et al., 2011) and TAP (30%, Ogren et al., 2017). These total measurement uncertainties are propagated according to appendix A of Perim De Faria et
al., 2021 to give an uncertainty for $\omega 0_{nt}$ (equation C4).

$$\Delta\omega = \sqrt{\left(\frac{\sigma_{sc}}{(\sigma_{sc} + \sigma_a)^2} \cdot \Delta\sigma_{sc}\right)^2 + \left(\frac{\sigma_a}{(\sigma_{sc} + \sigma_a)^2} \cdot \Delta\sigma_a\right)^2}$$

Equation C4: Error propagation for $\omega 0_{nt}$, where $\sigma_{sc}$ is independent scattering and $\sigma_a$ is independent absorption coefficients.

$\omega 0$ is not very sensitive to the real part of the index of refraction, and as such the real part of the estimated index
of refraction is not very well constrained (Peers et al., 2019). Figure C1 shows $\omega 0_{psd}$ derived using IOR=1.615+0.012j and IOR=1.59+0.012j both yield a mean $\omega 0_{psd}$ of 0.917. As such, we use a real aspect of 1.59 as derived by McMeeking et al., 2012 during their airborne measurement campaign over London, UK in 2009. Where insufficient data is available to enable calculation of the $\omega 0$ and thus IOR, an IOR for flights in a similar location and meteorological conditions is adopted. The uncertainties associated with applying a flight-
mean IOR is investigated in more depth in the following case study.

**Case study**

Section 2.7 describes the processing applied to particle sizing measurements to account for sizing errors caused by differences in the IOR between the calibrant and ambient particles. The method applies corrections based on the assumption of a single ambient IOR per flight, which was derived via an iterative process based on
achieving closure with independent observations of particles single scattering albedo. In this section we undertake a sensitivity study to evaluate the magnitude of error arising from the assumption of a flight-mean IOR, based on variability observed during an example flight: M270, a high-Density Plume Mapping sortie north of Cambridge, where a sequence of straight and level runs at altitudes from 0.30 to 1.32 km were performed (Fig C2 and table C1). The wide range of altitudes over a single flight allows examination of the impact of a
potentially changing airmass with altitude on derivation of a flight mean IOR. Refer to Sect 4.3 for a description of meteorological conditions for this flight.

The range of measured single scattering albedos, $\omega 0_{nt}$ during flight M270 varied throughout the boundary layer (0.886 to 0.944, Fig C1 red crosses) and yielded a flight mean $\omega 0_{nt}$=0.921 ± 0.019σ (Fig C1, red line). These values fall within the range of single scattering albedo's observed by McMeeking et al., 2011 during airborne



observations over London (typically from 0.85 in urban plumes to 0.95 in regional pollution and background aerosol).

A flight mean $\omega0_{psd}$=0.917±0.10 σ (Fig C1, blue line) was calculated using a particle size distribution (PSD) corrected with an optimally derived IOR=1.59+0.12j (herein referred to as $IOR_{DER}$). To examine sensitivity in particle sizing due to variability in observed ω0 throughout the column, we also undertook PSD corrections

based on achieving closure between $\omega0_{psd}$ and the maximum observed $\omega0_{nt}$ ($IOR_{MAX}$, 1.59+0.008j), minimum $\omega0_{nt}$ ($IOR_{MIN}$, 1.59+0.016j) and an uncorrected PSD (retains the calibrant (PSL) IOR; $IOR_{PSL}$, 1.615+0.001j), shown as the grey dotted, dashed and dash-dot lines, respectively, on Fig C1.

Regression analysis (Fig C3, left column) of normalised PSD's corrected to $IOR_{MIN}$ (top) $IOR_{MAX}$ (middle) and $IOR_{PSL}$ (bottom) against $IOR_{DER}$ show good agreement, with $r^2$ of 0.9998, 0.9980 and 0.9983, respectively.

Mean differences between $IOR_{MIN}$:$IOR_{DER}$ , $IOR_{MAX}$:$IOR_{DER}$  and $IOR_{PSL}$:$IOR_{DER}$ (Fig C3, right column) are 9%, 10% and 23%, respectively. The comparatively large uncertainty between corrected and uncorrected size distributions underlines the importance of accounting for IOR corrections when making ambient aerosol measurements. Mean differences in all comparisons are largest where Dp ≈> 0.4 μm (PSD bin 15). Particle sizes in this region are comparable to the wavelength of light of the POPS (405 nm), which are the most efficient at

scattering shortwave radiation and sizes larger than this can be influenced by Mie resonances (Liu and Daum, 2000).

Flight M270 was chosen based on it showing significant variability compared to other Clean Air flights; uncertainty in using a flight-mean IOR for less varying flights is expected to be less. For example, flight M302, a typical London survey on 22nd July 2021, performed numerous runs at altitudes≈0.5km and yields a difference

of <2% between distributions corrected by $IOR_{MIN}$ and $IOR_{MAX}$.

In summary, we conclude that use of a flight-mean IOR approach in correcting size distribution data introduces modest uncertainty of <10% compared to applying a variable IOR approach.

**Appendix D: PM2.5 composition and density**

 **Appendix E: AURN sites used in section 4.4, Long term observations over London**

**Appendix F: NetCDF variables**





**Figures**

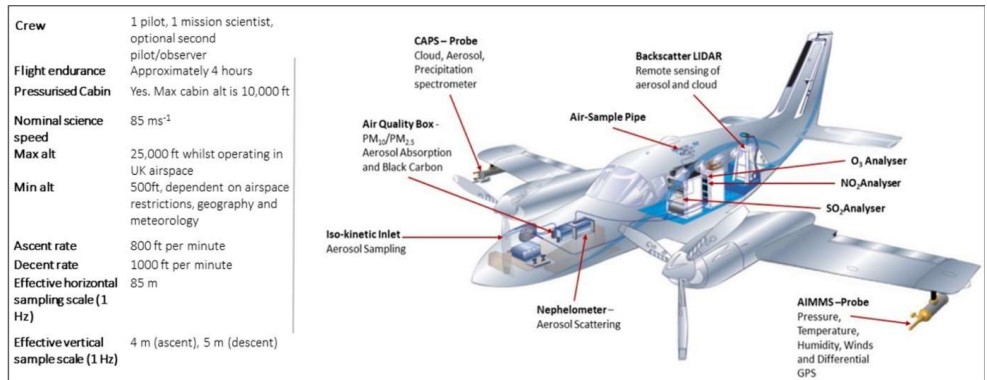

**Figure 1: The Met Office Atmospheric Survey Aircraft with instrumentation. Image courtesy Debbie**

**O'Sullivan, Met Office, 2021.**

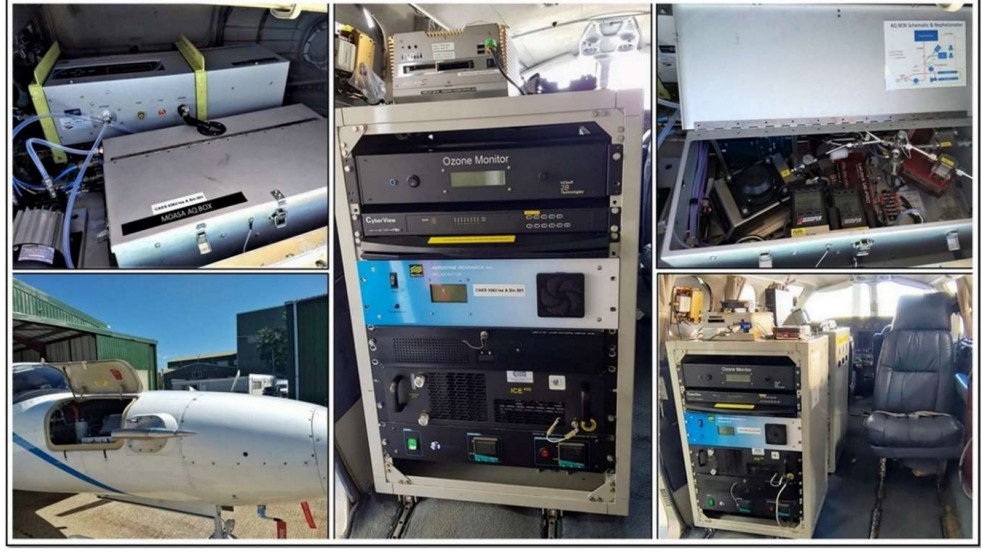

**Figure 2: Clockwise starting top left: the AQ box (foreground) and nephelometer (background) in the**

**MOASA nose bay; the aft instrumented rack housing the O₃, NO₂ and aerosol LIDAR control system;**

**inside the AQ box; inside the cabin looking forward; the Brechtel isokinetic air sample inlet and nose bay**

**of the MOASA.**



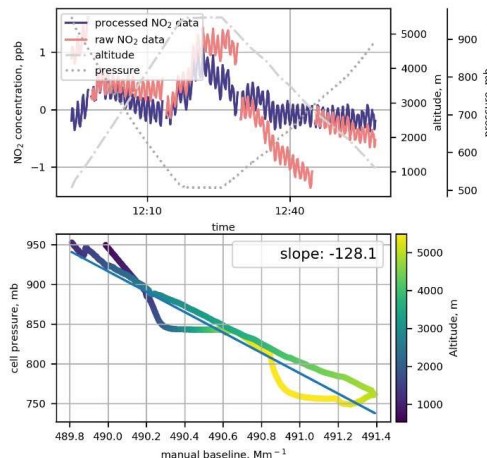

**Figure 3: Top: timeseries of unprocessed and processed NO₂ concentration. Bottom: baseline against cell pressure, coloured by altitude, with a linear fit shown with a blue line. Data obtained during flight M304 in November 2021 and averaged over 10 second intervals.**

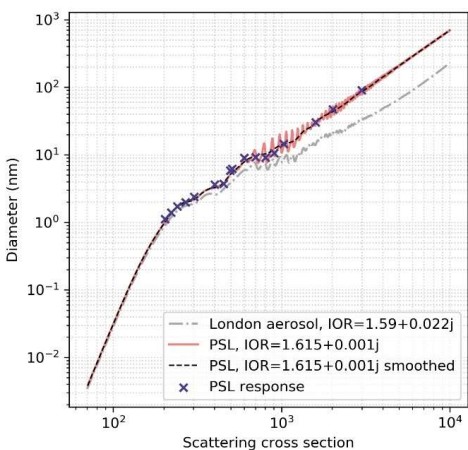


**Figure 4: Theoretical MOASA POPS Mie responses for PSL calibrant (1.615+0.001j) and ambient aerosol over London: 1.59-0.022j (McMeeking et al., 2012). Crosses are PSL responses from calibration on 16[th] September 2021.**



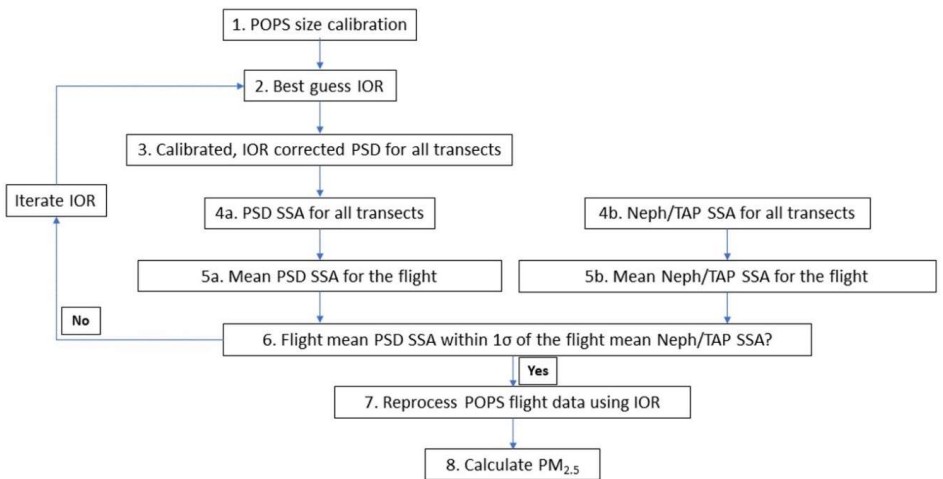

**Figure 5: Process to estimate the IOR of the ambient sample by iteratively adjusting the index of refraction of the POPS size distribution measurements until the POPS single scattering albedo matches the single scattering albedo from the nephelometer and TAP.**

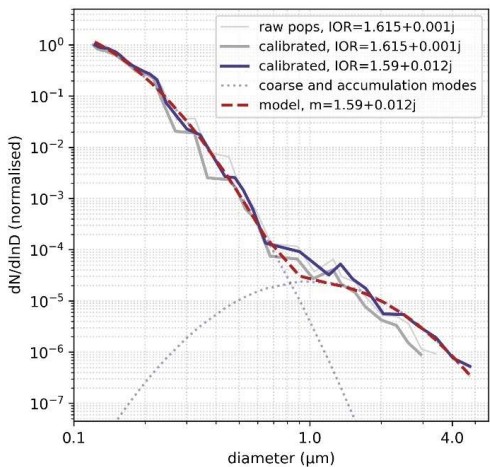

    **Figure 6: An example of raw, calibrated and calibrated with IOR-correction (IOR=1.59+0.12j) particle**
**size distributions, where the Y axis is normalised to 1. Overlaid are lognormal accumulation and coarse modes (dotted) plus the combination of these lognormal modes (dashed) fitted to the calibrated with IOR correction (blue solid line) size distribution.**

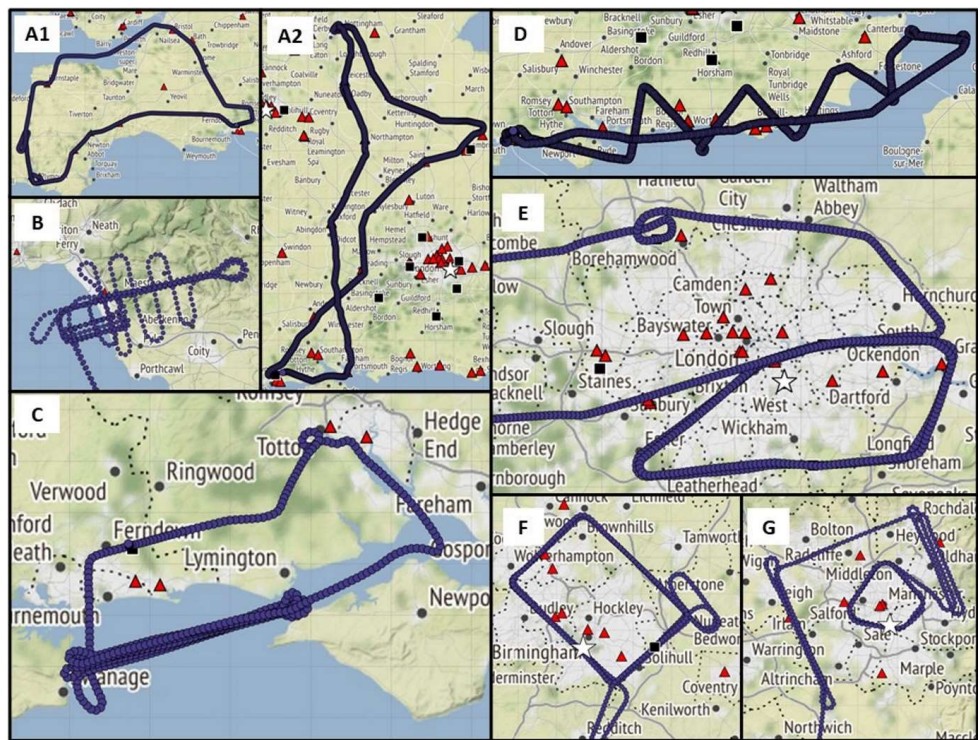

**Figure 7: Aircraft flight tracks for a typical (A) ground network survey over the south west (A1) and east (A2), during M288 and M262 on 19th May 2021 and 10th January 2020, respectively, (B) high density vertical mapping over Port Talbot, South Wales, during M284 on 24th March 2021, (C) south coast survey flight, during M301 on 27th July 2021, with focus on overpasses of the Solent and Southampton water, (D) coastal transition flight, during M285 on 30th March 2021, (E) London city survey flight IOP, M297 on 2nd July 2021. (F) Birmingham IOP flight (left), during M296 on 1st July 2021, (G) a typical Manchester IOP flight, during M300 on 20th July 2021. AURN sites are shown as triangles, airports as squares, stars are ground based supersites in Birmingham, Manchester and London. The geographical location of each sortie is shown in figure 8. Map tiles by Stamen Design, under CC BY 3.0. Data by OpenStreetMap, under ODbL.**

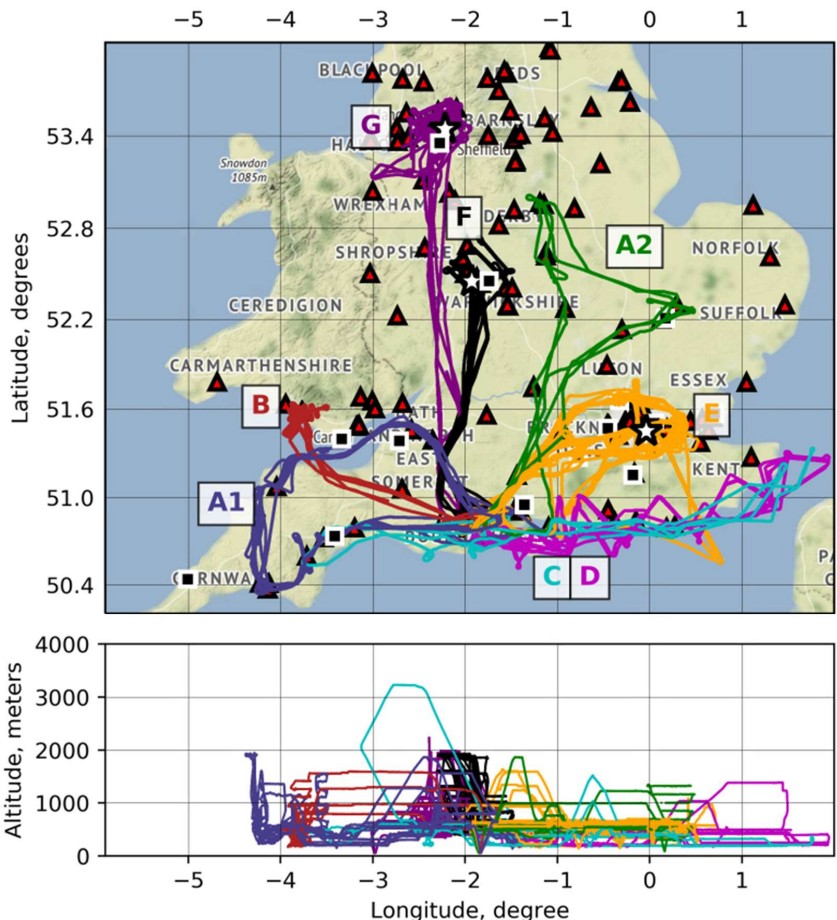

**Figure 8: Horizontal (top) and vertical (bottom) spatial coverage of 45 MOASA Clean Air flights from 27/07/2019 (flight M247) to 11/04/2021 (flight M299). AURN sites are shown as triangles, airports as squares, stars are ground based supersites in Birmingham, Manchester and London. The annotations relate to the sortie type detailed in Fig.7 where A1 and A2 are Ground Network Surveys, B are High Density Plume Mapping flights, C are South Coast Surveys, D are Coastal Transition Surveys, E are London City Surveys and F and G are the Birmingham and Manchester, respectively, IOP flights. Map by Stamen Design, under CC BY 3.0. Data by OpenStreetMap, under ODbL.**





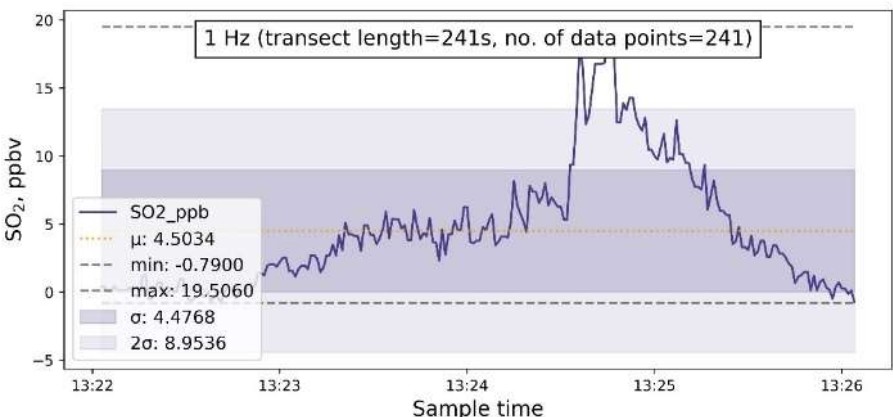

Figure 9: SO₂ timeseries from 13:22:30 – 13:26:50 during high density mapping flight M284.

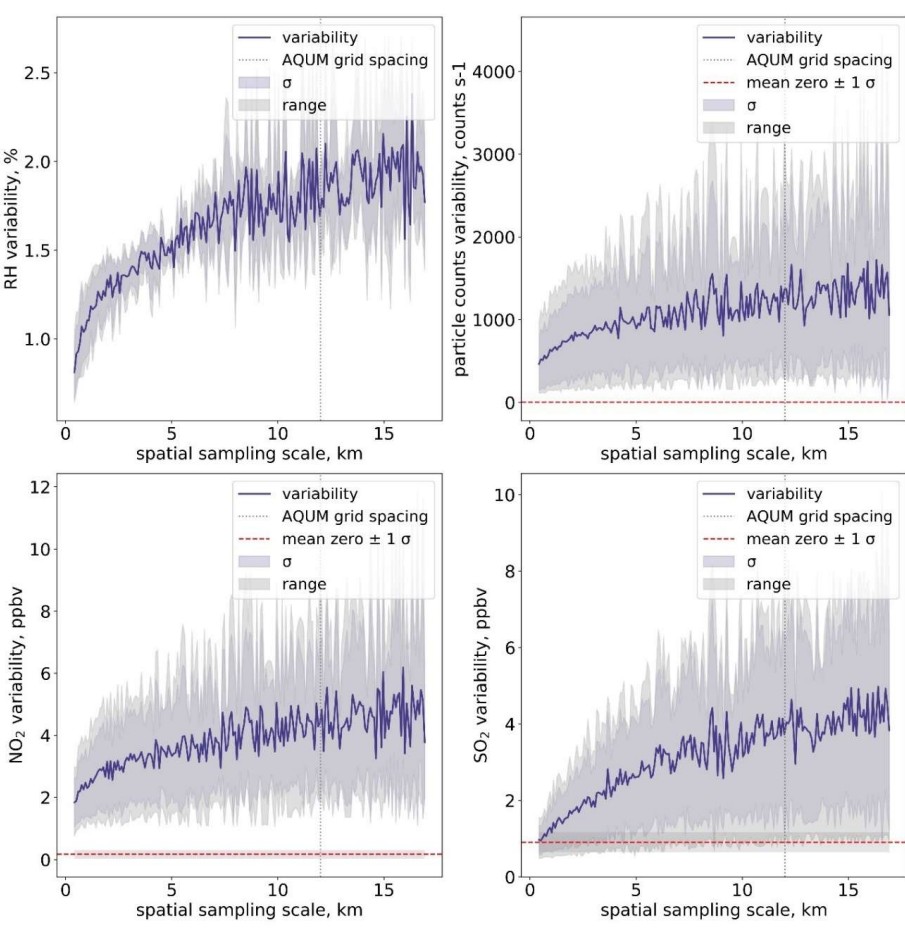





**Figure 10: Mean standard deviation (std, blue line) of relative humidity, particle counts, NO₂ and SO₂
over flight M284, as a function of spatial scale from 0.85 km to 17 km. The horizontal dashed red lines
represent instrument precision (±1 σ) derived from ground-based zero tests (where available).**

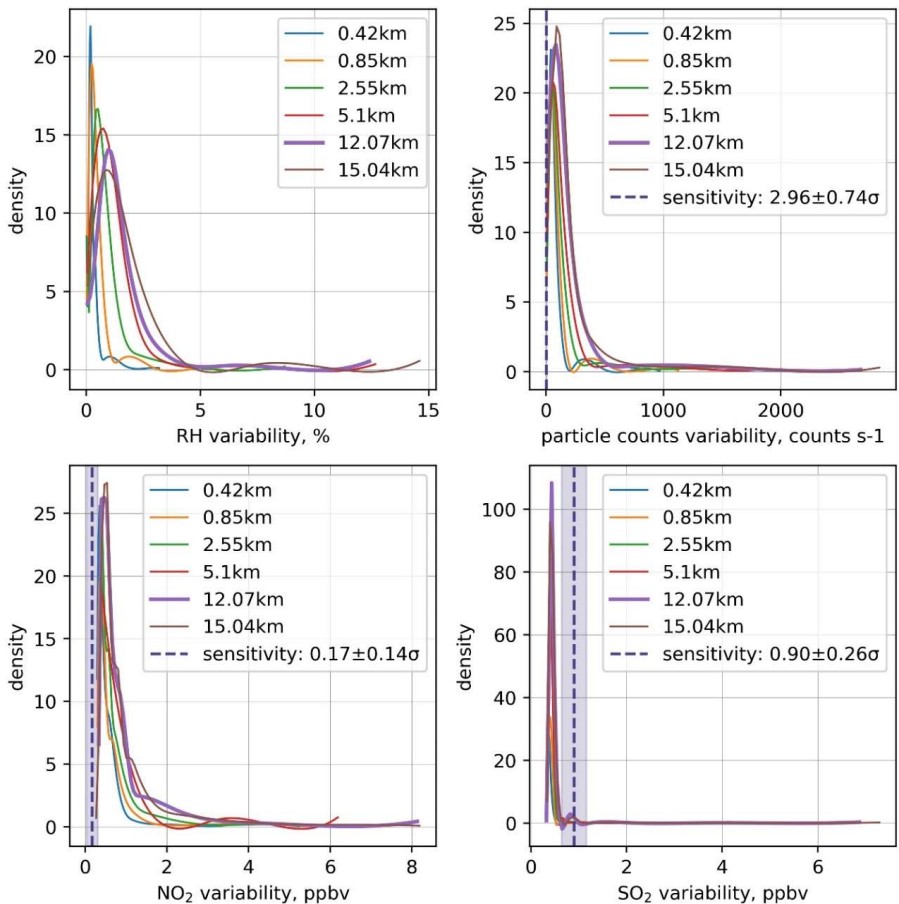

**Figure 11: Density distributions of RH, particle counts, NO₂ and SO₂ variability over the Clean Air
campaign, for $d_{int}$= 0.85, 0.42, 0.85, 2.55, 5.10, 12.07, and 15.04 km.**



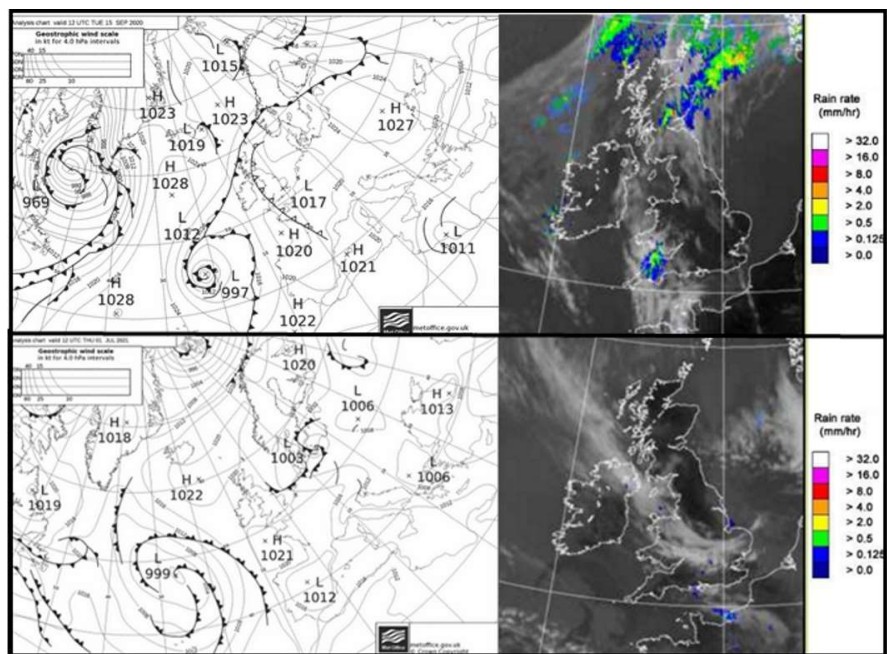

**Figure 12: Met Office synoptic chart and combined infra-red and rain-radar images for 12:00 UTC 15th September 2020 (top) and 1st July 2021 (bottom) (The National Meteorological Library, 2020).**

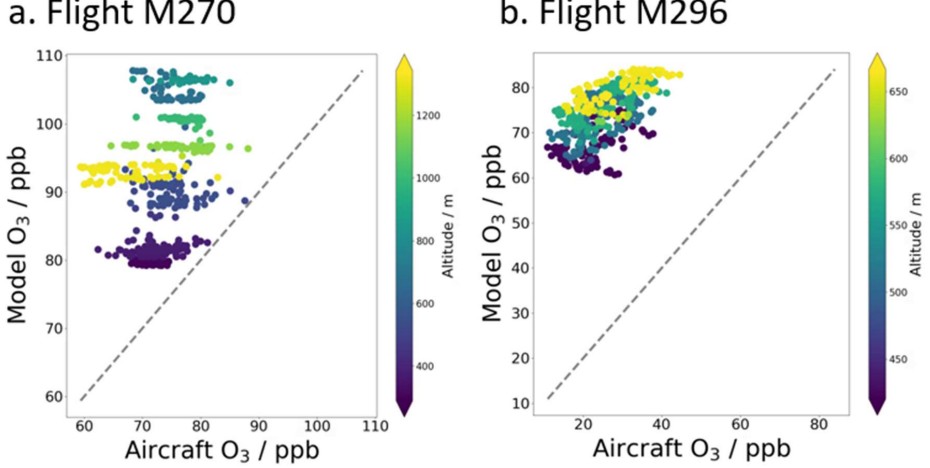

**Figure 13: Correlation of model and aircraft O$_3$ concentrations. Data averaged over 10 second intervals. Markers coloured by altitude. Dashed grey line represents agreement between the two datasets. Data shown for (a) Flight M270 on 15th September 2020, from 12:13:00 to 13:38:00 (the duration of the stacked level runs north of Cambridge) and (b) Flight M296 on 1st July 2021 from 11:23:00 to 12:52:00 (the duration of the Birmingham city circuits).**



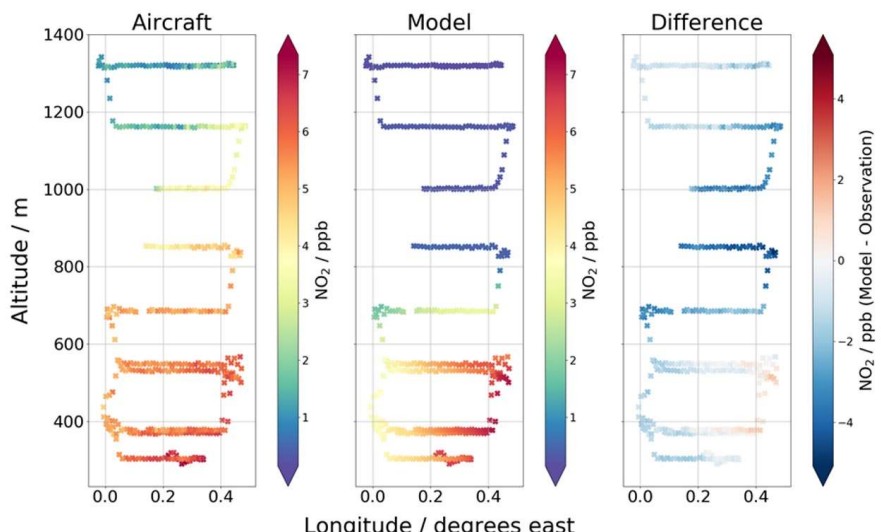


**Figure 14: Longitude-altitude plot of NO₂ concentration for vertically stacked transects during flight M270 on 15ᵗʰ September 2020. The left-hand figure shows the aircraft data, the middle figure shows the model data, and the right-hand figure shows the difference between the model and aircraft. Data averaged over 10 second intervals.**

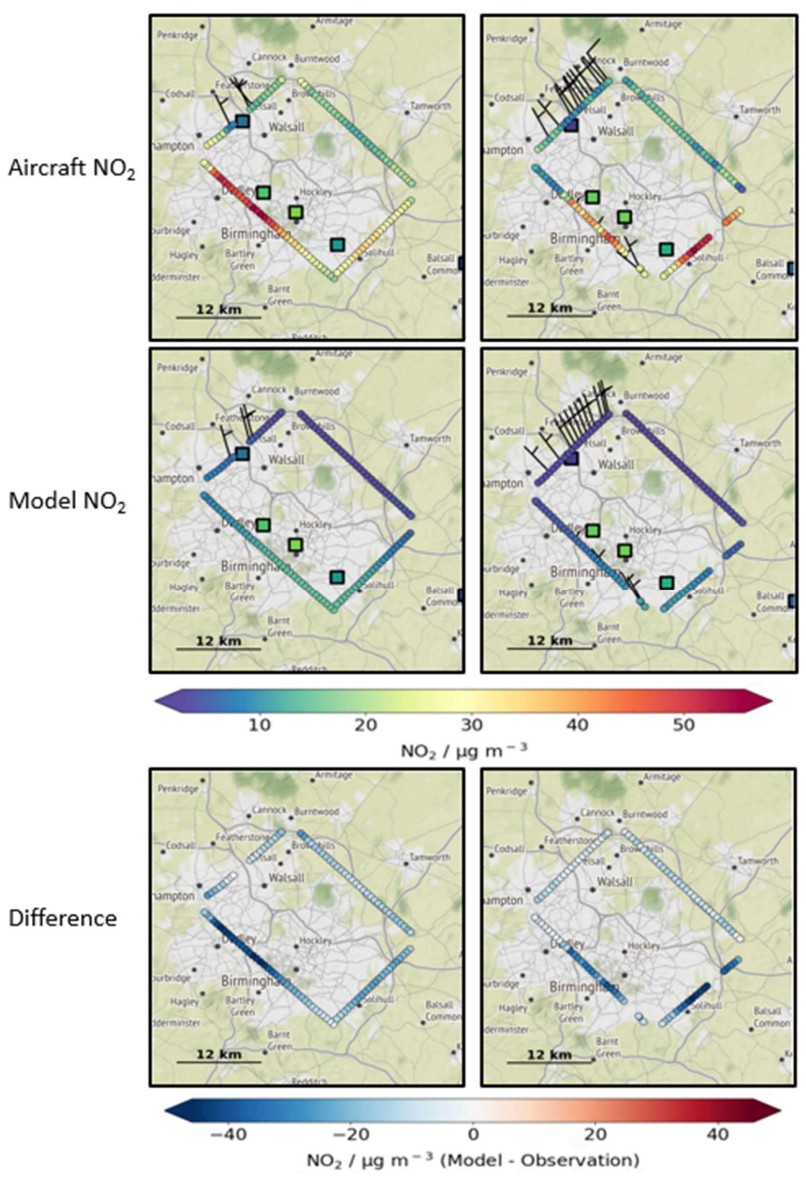


**Figure 15:** Aircraft flight tracks coloured by NO₂ concentration (μg/m³) for the first (left, 11:23 to 11:43) and fourth (right, 12:33 to 12:52) circuit, at altitudes of 423 and 657 metres, respectively, around Birmingham during flight M296 on 1ˢᵗ July 2021. Top row shows the aircraft data, middle row shows the model data and bottom row shows the difference between the model and observations. Observation data

is from straight and wings level transects and all data is averaged over 10 second intervals. Wind barbs are only shown where the observed wind components exceed the measurement uncertainty. Data in boxes is the hourly surface level AURN NO₂ concentration for the circuit. Map tiles by Stamen Design, under CC BY 3.0. Data by OpenStreetMap, under ODbL.



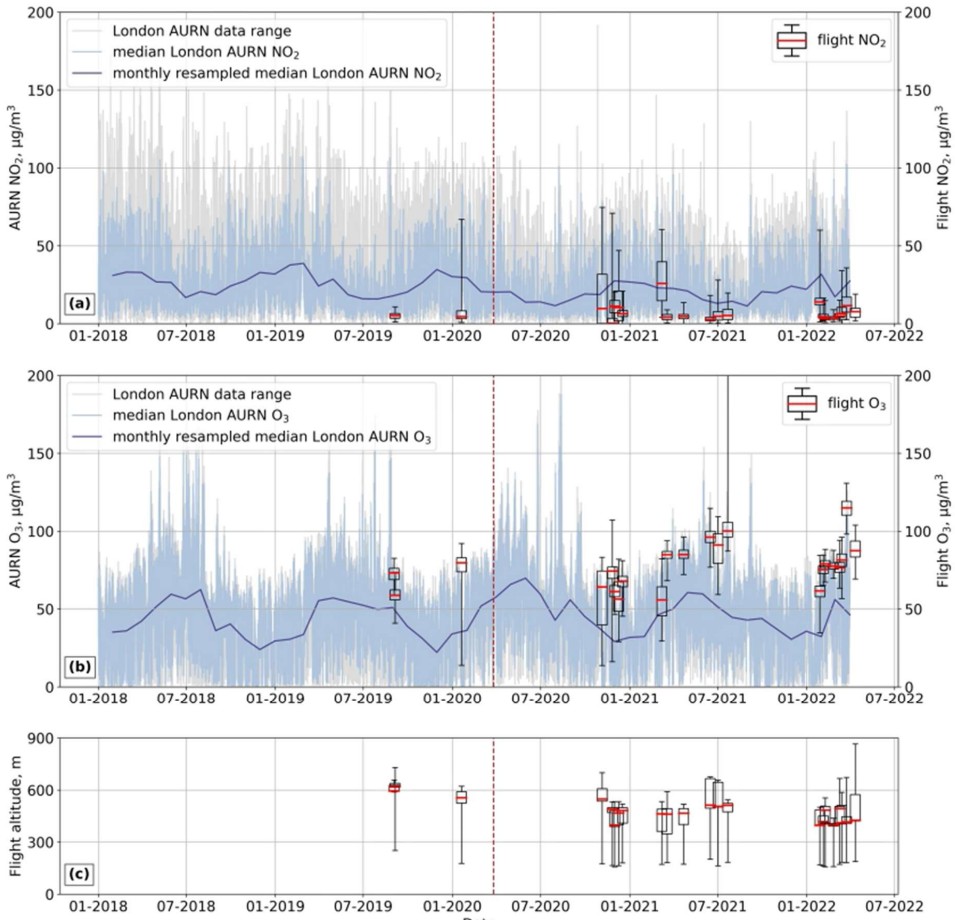

**Figure 16: AURN surface-level NO₂ (a) and O₃ (b) from all urban, rural and industrial background sites within the Greater London region, from February 2018 to end of March 2022. The monthly resampled median value is shown in dark blue. Also shown are box and whisker plots of NO₂ and O₃ aircraft observations at concurrent dates, where the box extends from the 25th to 75th percentiles, the red horizontal lines show the median and the whiskers show the range of data. Panel (c) shows the corresponding aircraft altitude, with the box-and-whisker plots retaining the afore-mentioned conditions. The vertical dashed red line in (a), (b) and (c) indicates the start of the COVID-19 impacted period, on 26th March 2020.**



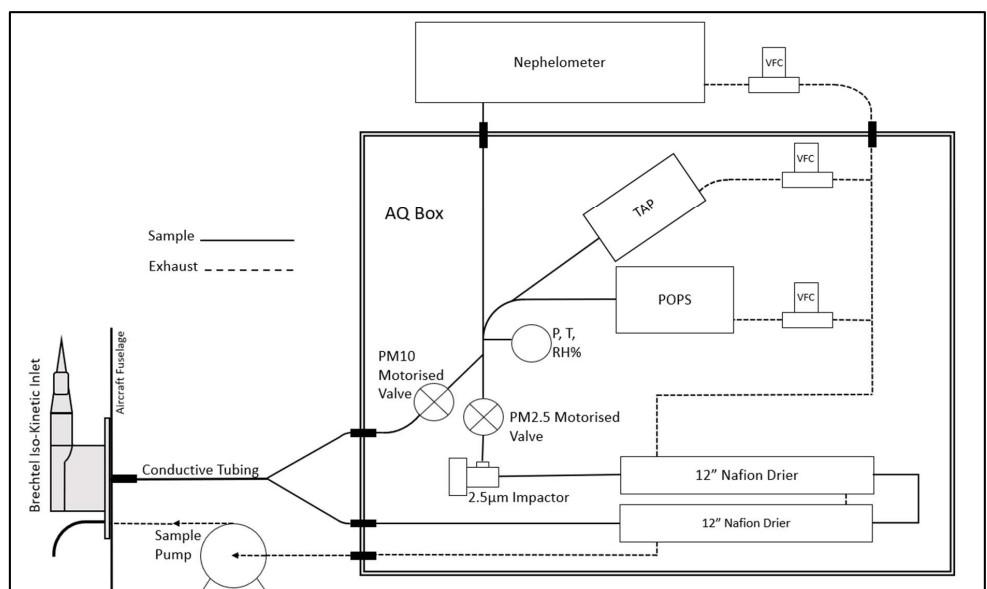

**Figure A1: Air Quality box flow schematic.**

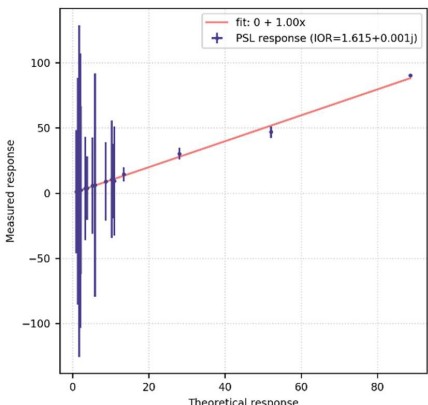

**Figure B1: POPS calibration from 16th September 2021. The blue circles represent PSL calibration beads with nominal diameters from 200 to 3000 nm. The vertical bars represent the error in response for each bead size and is the mean standard error of the mean for 15 second segments of each bead response.**

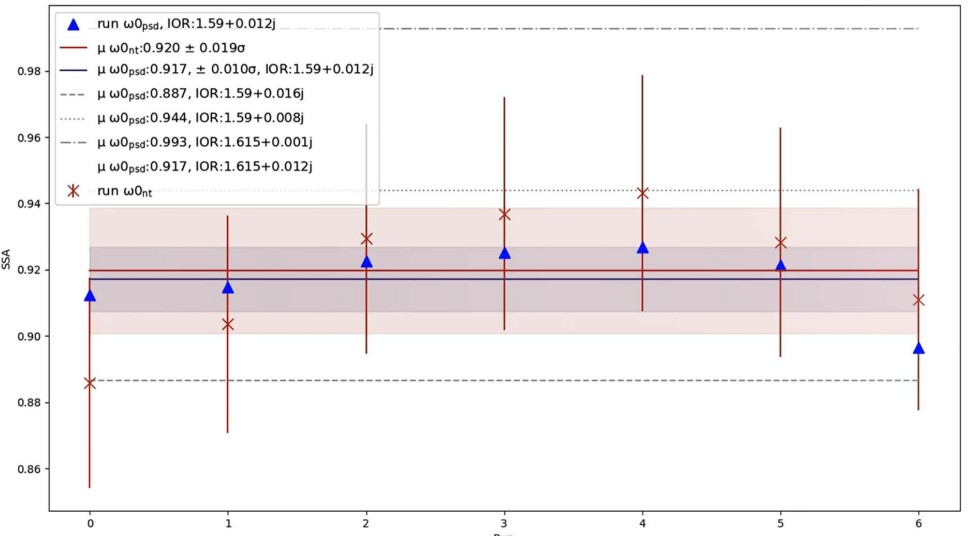


**Figure C1: Empirically derived nephelometer and TAP single scattering albedo (ω0$_{nt}$, red, crosses) and theoretically derived particle size distribution single scattering albedo (ω0$_{psd}$, blue, triangles) for 7 straight and level runs for flight M270 on 15$^{th}$ September 2021 north of Cambridge. Flight mean ω0$_{nt}$ and ω0$_{psd}$ with 1 σ variance (solid lines and shaded areas in red and blue, respectively) are shown. Also shown are the mean ω0$_{psd}$ derived using particle size distributions (PSD) corrected with the IOR which yielded ω0$_{psd}$ that closely matches the minimum ω0$_{nt}$ (run 0, grey dashed line) and maximum ω0$_{nt}$ (run 4, grey dotted line), where PSD IOR = 1.59+0.016j and 1.59+0.008j, respectively. The mean ω0$_{psd}$ derived using uncorrected (PSL-calibrant IOR=1.615+0.001j) PSD's is also shown (grey dot-dash line). The mean ω0$_{psd}$ derived using a real component of 1.615 and imaginary component of the retrieved IOR (0.12) is detailed in the legend (line not shown).**

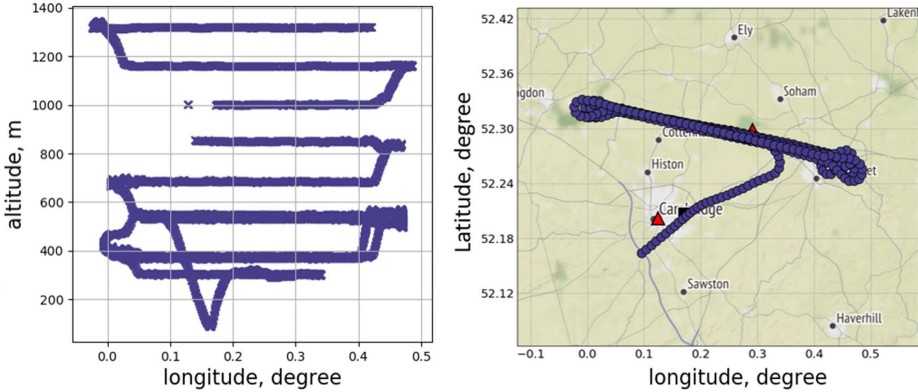

**Figure C2: MOASA flight track for M270 north of Cambridge on 15$^{th}$ September 2020 in the vertical (left) and horizontal (right). Triangles are AURN sites, the square is Cambridge airport. Map tile by Stamen Design, under CC BY 3.0. Data by OpenStreetMap, under ODbL.**





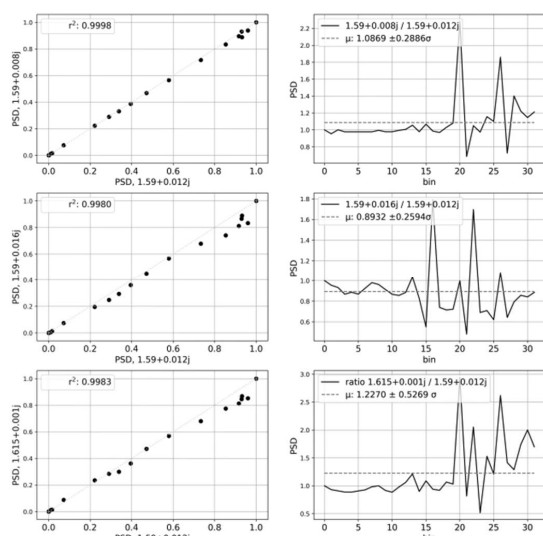


**Figure C3: Regression analysis (left) of flight M270 run 0 normalised particle size distribution (PSD) derived using IOR=1.59+0.016j (IOR$_{MIN}$) against PSD derived from IOR=1.59+0.008j (IOR$_{MAX}$, top) and 1.615+0.001j (IOR$_{PSL}$) against 1.59+0.012j (IOR$_{DER}$, bottom). Corresponding ratios of the same PSD's are to the right.**

**Tables**

| Sortie Type | Number flown | Flight numbers (number of designated runs in flight) |
|---|---|---|
| Southwest Ground Network Survey | 7 | M247 (4), M256 (3), M263 (5), M266 (3), M267 (5), M286 (7), M288 (4) |
| Northeast Ground Network Survey | 2 | M253 (4), M262 (3) |
| South Coast Survey | 5 | M250, M258 (5), M265 (4), M269 (4), M301 (6) |
| Coastal Transition | 6 | M272 (3), M280 (9), M283** (N/A), M285 (11), M289 (9), M322 (N/A) |
| High Density Spatial Mapping | 5 | M257 (2), M270, Cambridge (7), M274, Dover straights (4), M281, Port Talbot (10), M284, Port Talbot (4) |
| London | 23 | M251$^{NO2}$ (5), M252 (3), M264 (3), M273 (4), M275* (3), M276* (4), M277* (5), M278* (4), M279* (3), M282* (6), M287* (4), M294*$^{iop}$ (5), M297*$^{iop}$ (9), M302* (6), M305**$^{NO2O3}$ (N/A), M311*$^{iop}$ (4), M314*$^{iop}$ (5), M315*$^{iop}$ (5), M319*$^{iop}$ (6), M323* (5), M324* (5), M325* (4), M326* (4) |
| Birmingham IOP | 8 | M290 (8), M291 (9), M295 (10), M296 (9), M310 (7), M312 (12), M313 (12), M316 (N/A), |
| Manchester IOP | 7 | M292 (7), M293 (7), M298 (5), M299 (6), M300 (6), M317 (5), |





| | | M320 (6) |
|---|---|---|
| **Total flights** | **63** | |

**Table 1: MOASA Clean Air flights by sortie. The numbers in brackets indicate the number of straight and level transects used to derive the index of refraction for PM$_{2.5}$ (where applicable) and (from flights M247 to M302) the analysis in Sect 4.1. "N/A" indicates that no runs were used in forward analysis. London flights which include a central overpass are postfixed with an asterisk. Flights with limited data are postfixed with a double asterisk. London flights during the ground based IOP's are also postfixed with superscript 'iop'. London flight with no NO$_2$ data or O$_3$ data are post fixed 'NO2' or 'O3' (applicable to Sect 4.4).**

| Run | Times | Mean altitude (m) | $\omega 0_{nt}$ | $\omega 0_{psd}$ (IOR = 1.59 + 0.012j |
|---|---|---|---|---|
| 0 | 12:16:20 – 12:20:20 | 304 | 0.886 ± 0.03 | 0.912 |
| 1 | 12:37:50 – 12:42:20 | 378 | 0.904 ± 0.03 | 0.915 |
| 2 | 12:52:30 – 12:57:50 | 686 | 0.929 ± 0.03 | 0.923 |
| 3 | 13:00:00 – 13:04:50 | 851 | 0.937 ± 0.04 | 0.925 |
| 4 | 13:08:40 – 13:13:10 | 1002 | 0.943 ± 0.04 | 0.927 |
| 5 | 13:16:20 – 13:21:10 | 1162 | 0.928 ± 0.03 | 0.921 |
| 6 | 13:23:50 – 13:29:20 | 1320 | 0.911 ± 0.03 | 0.897 |
| | Flight averages | 814.71 | 0.920 ± 0.019 σ | 0.917 ± 0.010 σ |

**Table C1: Mean altitude and single scattering albedo for seven runs during flight M270 on 15$^{th}$ September 2020.**

| | Weighted average density (gcm$^3$) | Total mass | Black carbon | Organic carbon | NH$_4$NO$_3$ & NaNO$_3$ | (NH$_4$)$_2$SO$_4$ | NaCl | CaSO$_4$ anhydrous | Fe-rich dust | Other (incl. bound water |
|---|---|---|---|---|---|---|---|---|---|---|
| Index of refraction | - | - | 1.95-0.79j [5] | 1.63-0.021j [1] | NH4NO3: 1.550, 0j [6] | 1.53+ 0j [1] | 1.54 + 0j [3] | 1.57 [8] | 2.80-3.34j (Iron) [3] | 1.33+ 0.0j |
| Density (g/cm$^3$) | - | - | 1.8 [1] | 1.35 [2] | 1.72 [2] | 1.77 [3] | 2.17 [3] | 2.96 [7] | 2.5 [4] | 1 |
| **Study** | | | **%** | **%** | **%** | **%** | **%** | **%** | **%** | **%** |
| H2004_ub | 1.69 | 100 | 14.4 | 25.1 | 14.6 | 21.3 | 2.1 | 1.9 | 10.2 | 10.4 |
| H2004_ubhp | 1.69 | 100 | 9.144 | 15.94 | 36.5 | 13.53 | 1.33 | 1.21 | 6.477 | 6.604 |





| | | | | | | | | | | |
|---|---|---|---|---|---|---|---|---|---|---|
| AG2012_ub | 1.71 | 100 | 11.81 | 20.59 | 29.94 | 17.47 | 1.72 | 1.56 | 8.37 | 8.53 |
| H2008_ab | 1.55 | 100 | - | 24-59 | 20 -39 | 21-37 | - | - | - | - |
| H2008_abmed | 1.58 | 100 | - | 41.5 | 29.5 | 29 | - | - | - | - |
| H2008_abmo | 1.51 | 100 | - | 59 | 20 | 21 | - | - | - | - |
| H2008_abmi | 1.65 | 100 | - | 24 | 39 | 37 | - | - | - | - |
| Mean Pp | 1.64 ± 0.07 (1σ) | | | | | | | | | |

**Table D1: Average chemical composition and density (Pp, g/cm3) of UK PM2.5. Where H2004_ub and H2004_ubhp are Harrison et al., 2004 urban background and urban background high pollution, respectively. High pollution percentages represent findings by Harrison et al., 2004, who reported an approximate doubling of concentrations of elemental carbon, organic compounds, sodium nitrate, ammonium sulphate, calcium sulphate and iron-rich dusts on high pollution days, and an increase of more than five-fold in the ammonium nitrate concentration. AG2012_ub: Air Quality Expert Group, 2012 urban background, H2008_ab is Haywood, 2008, airborne measurements derived from the Facility for Airborne Atmospheric Measurements Bae146 over 3 flights (shown as reference ranges only). H2008_abmed, H2008_abmo and H2008_abmi: median, maximum organics and maximum inorganics, respectively for H2008_ab percentage ranges. Index of Refraction and Density: The numbers in square brackets refer to the reference for the associated value, which are as follows: [1] Morgan et al., 2010, [2] Haywood, 2008, [3] Hinds, 1999, [4] Lafon et al., 2006. [5] Bond and Bergstrom, 2006, [6] Hoon Jung et al., 2016, [7] CAMEO chemicals, NOAA, n.d. [8] PubChem, n.d. An assumed density of 1 gcm$^3$ is used for `Other including bound water'.**

| NO$_2$ | Site type | Mean hourly concentration pre-lockdown* (µg/m$^3$ ± 1σ) | Mean hourly concentration post-lockdown** (µg/m$^3$ ± 1σ) |
|---|---|---|---|
| London Bexley | Suburban background | 22.16 ± 17.21 | 19.32 ± 16.13 |
| London Bloomsbury | Urban background | 32.73 ± 18.52 | 26.36 ± 16.81 |
| London Eltham | Suburban background | 16.33 ± 13.09 | 13.56 ± 10.92 |
| London Haringey Priory Park South | Urban background | 21.29 ± 15.44 | 17.02 ± 13.28 |
| London Hillingdon | Urban background | 43.73 ± 26.84 | 26.01 ± 17.83 |
| London N. Kensington | Urban background | 26.98 ± 18.21 | 19.68 ± 15.21 |
| London Westminster | Urban background | 31.64 ± 18.52 | 24.69 ± 16.44 |
| | **Mean** | **27.84 ± 18.26** | **20.95 ± 15.23** |
| **O$_3$** | **Site type** | **Mean hourly concentration pre-lockdown* (µg/m$^3$ ± 1σ)** | **Mean hourly concentration post-lockdown** (µg/m$^3$ ± 1σ)** |
| London Bloomsbury | Urban background | 37.13 ± 22.01 | 43.42 ± 22.64 |





| London Eltham | Suburban background | 45.32 ± 25.62 | 39.94 ± 23.99 |
|---|---|---|---|
| London Haringey Priory Park South | Urban background | 45.39 ± 26.12 | 49.50 ± 25.73 |
| London Hillingdon | Urban background | 31.37 ± 24.22 | 38.66 ± 23.90 |
| London N. Kensington | Urban background | 47.20 ± 26.48 | 51.66 ± 25.26 |
| | **Mean** | **41.28 ± 25.22** | **44.63 ± 24.30** |

**Table E1: NO₂ and O₃ AURN sites used in Sec 4.4, Long term observations over London. All figures rounded to two significant figures. * Pre-lockdown: 26th March 2018 to 25th March 2020. Post-lockdown: 26th March 2020 to 25th March 2022.**

| | Standard name Aa prefix of instrument name e.g. "NEPH_" indicates a housekeeping parameter. | Units, long name, frequency, and comments (where applicable). [ ] indicates a changeable parameter |
|---|---|---|
| Dimension | time | units: seconds since flight_date 00:00:00 long name: the time the measurement was taken timezone: UTC frequency: 1 Hz |
| AIMMS | latitude | units: degree north long name: aircraft latitude measured by the AIMMS frequency: 1 Hz |
| | longitude | units: degree east long name: aircraft longitude measured by the AIMMS, frequency: 1 Hz longitude |
| | altitude | units: m long name: aircraft GPS height measured by the AIMMS frequency: 1 Hz comment: nominally above sea level |
| | air_temperature | units: K long name: ambient air temperature measured by the AIMMS frequency: 1 Hz |
| | relative_humidity | units: % long name: A measurement of the water vapor that exists in a mixture of air and water vapor measured by the AIMMS frequency: 1 Hz |
| | air_pressure | units: hPa long name: ambient air pressure measured by the AIMMS frequency: 1 Hz |





| | wind_speed | units: m s-1 |
|---|---|---|
| | | long name: wind speed measured by the AIMMS |
| | | comment: Applicable only when wings are level and wind speed is above the wind component speed threshold. Hence the wind speed flag is derived from the wind flow N flag the wind flow E flag and the roll angle flag. See config.py for the min and max limits of these parameters. |
| | | frequency: 1 Hz |
| | wind_from_direction | units: degree |
| | | long name: wind direction measured by the AIMMS |
| | | comment: The direction the wind is blowing from. Applicable only when wings are level and wind speed is above the wind component speed threshold. Hence the wind direction flag is derived from the wind flow N flag the wind flow E flag and the roll angle flag. See config.py for the min and max limits of these individual parameters. |
| | | frequency: 1 Hz |
| | roll_angle | units: degree |
| | | long name: the rotation about the longitudinal axis of the aircraft measured by the AIMMS |
| | | comment: zero degree indicates the wings on a fixed-wing aircraft are level with the local horizontal plane. |
| | | frequency: 1 Hz |
| | | positive: right wing down roll angle |
| | pitch_angle | units: degree |
| | | long name: the angle between the longitudinal axis of the aircraft and the horizon measured by the AIMMS |
| | | comment: zero degree indicates the nose and tail of the aircraft are level with the local horizontal plane |
| | | frequency: 1 Hz |
| | | positive: nose up |
| | true_air_speed | units: m s-1 |
| | | long name: true air speed measured by the AIMMS |
| | | frequency: 1 Hz |
| | northward_wind | units: m s-1 |
| | | long name: wind flow vector north component measured by the AIMMS |
| | | frequency: 1 Hz |
| | eastward_wind | units: m s-1 |
| | | long name: wind flow vector east component measured by |



| | | |
|---|---|---|
| | | the AIMMS<br>frequency: 1 Hz |
| | yaw_angle | units: degree<br>long name: yaw angle as measured by the AIMMS<br>instrument<br>frequency: 1 Hz |
| | downward_air_velocity | units: m s-1<br>long name: vertical wind as measured by the AIMMS<br>instrument<br>comment: positive is down<br>frequency: 1 Hz |
| | sideslip_angle | units: degree<br>long name: angle of sideslip as measured by the AIMMS<br>instrument<br>frequency: 1 Hz |
| Nephelometer | forward_scattering_red | units: Mm-1,<br>long name: corrected red (635nm) scattering (by gas and<br>particles, with dark count subtracted) over 0 - 170 degrees,<br>smoothed to 15s, measured by the Nephelometer,<br>frequency: 1 Hz |
| | forward_scattering_green | units: Mm-1,<br>long name: corrected green (525nm) scattering (by gas and<br>particles, with dark count subtracted) over 0 - 170 degrees,<br>smoothed to 15s, measured by the Nephelometer,<br>frequency: 1 Hz, |
| | forward_scattering_blue | units: Mm-1,<br>long name: corrected blue (450nm) scattering (by gas and<br>particles, with dark count subtracted) over 0 - 170 degrees,<br>smoothed to 15s, measured by the Nephelometer,<br>frequency: 1 Hz, |
| | backscattering_red | units: Mm-1,<br>long name: corrected red (635nm) scattering coefficient for<br>backscatter (by gas and particles, with dark count<br>subtracted) over 90 - 170 degrees, smoothed to 15s,<br>measured by the Nephelometer,<br>frequency: 1 Hz, |
| | backscattering_green | units: Mm-1,<br>long name: corrected green (525nm) scattering coefficient<br>for backscatter (by gas and particles, with dark count<br>subtracted) over 90 - 170 degrees, smoothed to 15s, |





| | | |
|---|---|---|
| | | measured by the Nephelometer, frequency: 1 Hz |
| | backscattering_blue | units: Mm-1, long name: corrected blue (450nm) scattering coefficient for backscatter (by gas and particles, with dark count subtracted) over 90 - 170 degrees, smoothed to 15s, measured by the Nephelometer, frequency: 1 Hz |
| | scattering_correction_green | units: Mm-1 long name: scattering corrections applied to correct raw blue scattering data measured by the Nephelometer frequency: 1 Hz |
| | scattering_correction_blue | units: Mm-1 long name: scattering corrections applied to correct raw blue scattering data measured by the Nephelometer frequency: 1 Hz |
| | scattering_correction_red | units: Mm-1, long name: scattering corrections applied to correct raw red scattering data measured by the Nephelometer, frequency: 1 Hz |
| | aerosol_angstrom_exponent | units: Mm-1 long name: Angstrom exponent as an average of wavelength pair Angstrom exponents, measured by the Nephelometer frequency: 1 Hz |
| | NEPH_sample_temperature | units: degree Celcius long name: sample air temperature measured by the Nephelometer frequency: 1 Hz |
| | NEPH_cell_temperature | units: degree Celcius long name: cell temperature measured via a sensor mounted in the cell wall (near the light source) (for the Nephelometer) frequency: 1 Hz |
| | NEPH_RH | units: % long name: sample air relative humidity measured by the Nephelometer frequency: 1 Hz |
| | NEPH_pressure | units: hPa long name: barometric pressure in the cell measured by the |





| | | |
|---|---|---|
| | | Nephelometer |
| | | frequency: 1 Hz |
| | NEPH_flow | units: litre min-1 |
| | | long name: sample flow rate measured by the Nephelometer |
| | | frequency: 1 Hz |
| TAP | absorption_coefficient_blue | units: Mm-1 |
| | | long name: Corrected (Virkkula et al, 2010) blue (wavelength = 467nm) absorption coefficient, measured by TAP. |
| | | frequency: 1 Hz |
| | absorption_coefficient_green | units: Mm-1 |
| | | long name: Corrected (Virkkula et al, 2010) green (wavelength = 528nm) absorption coefficient, measured by TAP. |
| | | frequency: 1 Hz |
| | absorption_coefficient_red | units: Mm-1 |
| | | long name: Corrected (Virkkula et al, 2010) red (wavelength = 652nm) absorption coefficient, measured by TAP. |
| | | frequency: 1 Hz |
| | TAP_sample_flow | units: litre min-1 |
| | | long name: Sample flow for the TAP, as measured by TAP. |
| | | frequency: 1 Hz |
| | TAP_sample_air_temp | units: degree Celsius |
| | | long name: Sample flow for the TAP, as measured by TAP. |
| | | Frequency: 1 Hz |
| | TAP_case_temp | units: degree Celsius |
| | | long name: Case temperature for the TAP, as measured by TAP. |
| | | frequency: 1 Hz |
| POPS | mass_concentration_of_dried_pm2p5_aerosol_in_air | units: µg m-3 |
| | | long name: Mass concentration of dried pm2p5 aerosol in air for the POPS instrument |
| | | frequency: 1 Hz |
| | | comment: assumes homogeneous spherical particles and a density of [density] g/cm^3. |
| | | Calculated using index of refraction [IOR] corrected, calibrated mid-bin diameters from the POPS instrument. |





| | | Diameter range used in PM2.5 calculations: [lower bin] μm to [upper bin] μm. Sample is dried (relative humidity typically below 20%). |
|---|---|---|
| | number_concentration_of_aerosol | units: cm3<br>long name: number concentration of aerosol measured by the POPS instrument<br>frequency: 1 Hz |
| | aerosol_particle_counts | units: counts s-1<br>long name: particle counts measured by the POPS instrument<br>frequency: 1 Hz |
| | dried_aerosol_size_spectra | units: counts<br>long name: number of dried particle counts per bin for the POPS instrument<br>frequency: 1 Hz |
| | bin_boundaries | units: nm,<br>long name: nominal and calibrated bin boundaries in terms of scattering cross section, and nominal, calibrated and calibrated IOR corrected bin boundaries in terms of diameter for the POPS instrument. See comment for key.,<br>frequency: N/A,<br>comment: Index for rows (1 to 16) are:<br>lr_ss, ur_ss, mbr_ss, mbr_ss_err, lr_d, ur_d, mbr_d, mbr_d_err, lc_ss, uc_ss, mbc_ss, mbc_ss_err, lc_d, uc_d, mbc_d, mbc_d_err where l=lower, u=upper, mb=mid-bin, ss=scattering_signal, d=diameter, r=raw, c=calibrated, err=error} |
| | POPS_sample_flow | units: cm3 s-1<br>long name: sample flow rate measured by the POPS instrument<br>frequency: 1 Hz<br>comment: measured by the laminar flow element and differential pressure sensor on the POPS instrument, |
| | POPS_bl | unit: counts<br>long name: baseline of the detector (raw analog-to-digital counts) measured by the POPS instrument<br>frequency: 1 Hz |
| | POPS_blth | units: N/A<br>long name: baseline threshold for particle counting<br>frequency: 1 Hz |





| | POPS_laserdiode_temp | units: degree Celcius <br> long name: temperature of laser diode control board measured by the POPS instrument <br> frequency: 1 Hz |
|---|---|---|
| | POPS_ld_mon | unit: arbitrary value <br> long name: laser diode output power monitor measured by the POPS instrument <br> frequency: 1 Hz |
| | POPS_laserfb | unit: arbitrary value <br> long name: feedback value used when controlling laser power using PID control measured by the POPS instrument <br> frequency: 1 Hz |
| | POPS_ambient_pressure | units: hPa <br> long name: ambient pressure as measured by the POPS instrument <br> frequency: 1 Hz |
| | POPS_pumpfb | unit: arbitrary value <br> long name: feedback value used when controlling pump speed using PID control measured by the POPS instrument <br> frequency: 1 Hz |
| | POPS_onboard_temp | units: degree Celcius <br> long name: on-board temperature measured by the POPS instrument <br> frequency: 1 Hz |
| NO2 | mass_concentration_of_nitrogen_dioxide_in_air | units: µ m-3 <br> long name: manually calculated nitrogen dioxide concentration with manual baseline subtracted, as measured by the NO2 instrument <br> frequency: 1 Hz |
| | concentration_of_nitrogen_dioxide_in_air | units: ppbv <br> long name': manually calculated nitrogen dioxide concentration with manual baseline subtracted, as measured by the NO2 instrument <br> frequency: 1 Hz |
| | NO2_manual_baseline | units: Mm-1 <br> long name: baseline used in manual concentration calculation of NO2. <br> frequency: 1 Hz |
| | NO2_man_baseline_1 | units: Mm-1 <br> long name: manually calculated baseline method 1 (for the |



| | | |
|---|---|---|
| | | NO2 instrument) |
| | | comment: derived by linearly interpolating between consecutive baseline measurements. |
| | | frequency: 1 Hz |
| | NO2_man_baseline_2 | units: Mm-1 |
| | | long name: manually calculated baseline method 2 (for the NO2 instrument) |
| | | comment: derived using defined scaling of Rayleigh corrected baseline loss (Mm-1) per mb to determine baseline based on NO2 cell pressure |
| | | frequency: 1 Hz |
| | NO2_man_baseline_3 | units: Mm-1 |
| | | long name: manually calculated baseline method 3, as per method 2, but do a linear fit to pressure-dependence of background only when pressure span is bigger than 250 mb (for the NO2 instrument) |
| | | frequency: 1 Hz |
| | NO2_cell_pressure | units: hPa |
| | | long name: cell pressure measured by the NO2 instrument |
| | | frequency: 1 Hz |
| | NO2_cell_temperature | units: K |
| | | long name: cell temperature measured by the NO2 instrument |
| | | frequency: 1 Hz |
| | NO2_N | units: cm-3 |
| | | long name: nitrogen dioxide molecular number density calculated at sensor pressure and temperature (for the NO2 instrument) |
| | | frequency: 1 Hz |
| O3 | mass_concentration_of_ozone_in_air | units: µg m-3, |
| | | long name: mass concentration of ozone measured by the ozone instrument |
| | | frequency: 0.5 Hz |
| | concentration_of_ozone_in_air | units: ppbv |
| | | long name: concentration of ozone measured by the ozone instrument |
| | | frequency: 0.5 Hz |
| | O3_cell_pressure | units: hPa |
| | | long name: measurement-cell pressure measured by the ozone instrument |





| | | frequency: 0.5 Hz |
| --- | --- | --- |
| | O3_cell_temperature | units: degree_Celsius |
| | | long name: measurement-cell temperature measured by the ozone instrument |
| | | frequency: 0.5 Hz |
| | O3_volumetric_flow_rate | units: cm3 min-1 |
| | | long name: volumetric flow rate measured by the ozone instrument |
| | | frequency: 0.5 Hz |
| SO$_2$ | mass_concentration_of_sulfur_dioxide_in_air | units: µg m-3 |
| | | long name: mass_concentration_of_sulfur_dioxide_measured_by_the_SO2_instrument |
| | | frequency: 1 Hz |
| | | comment: minimum detection limit ± 2.661 µg m-3. Be aware of signal to noise at low concentrations |
| | concentration_of_sulfur_dioxide_in_air | units: ppbv |
| | | long name: sulfur dioxide concentration measured by the SO2 instrument |
| | | frequency: 1 Hz |
| | | comment: minimum detection limit ± 1PPB. Be aware of signal to noise at low concentrations |
| | SO2_internal_temperature | units: degree_Celsius |
| | | long name: internal temperature measured by the SO2 instrument |
| | | frequency: 1 Hz |
| | SO2_reaction_temperature | units: degree_Celsius |
| | | long name: reaction (or chamber) temperature measured by the SO2 instrument |
| | | frequency: 1 Hz |
| | SO2_pressure | units: hPa |
| | | long name: reaction chamber pressure measured by the SO2 instrument |
| | | frequency: 1 Hz |
| | SO2_flow | units: litre min-1 |
| | | long name: sample flow measured by the SO2 instrument |
| | | frequency: 1 Hz |

1170 **Table F1: Units, description (long name) and frequency of Clean Air NetCDF variables.**