# Peer review of "Long-term airborne measurements of pollutants over the UK, including during the COVID-19 pandemic, to support air quality model development and evaluation"

_Atmospheric Measurement Techniques, 2023_

## Referee Comment (RC1)

**Referee comment on AMT-2023-15**

The manuscript entitled "Long-term airborne measurements of pollutants over the UK, including during the COVID-19 pandemic, to support air quality model development and evaluation" by Angela Mynard et al., introduces a dataset of airborne observations taken with the MOASA measurement platform during 63 flights over UK in the period June 2019 to April 2022. These are useful and interesting measurements that should be shared with the scientific community. However, while the measurements likely provide important new information, the manuscript has the character of a technical report and the analysis of the few data examples presented is poor and needs to be extended.

The manuscript can be published in AMT after a major revision of the structure and the scientific contribution of the work.

**General comments**

Apart from the potential usefulness of the dataset, the conclusions of the paper are unfortunately lost in the extended technical description of the dataset. The interpretation of the examples presented is generally too simplistic. The manuscript would benefit from moving most of the technical details of the database to the supplement while extending the analysis of the examples given to provide a concrete picture of the actual improvement achieved through this observations in air quality model development and evaluation, in particular for the case of the AQUM regional model, as indicated by the title and announced in the introduction.

In addition, the interpretation of the differences in ground based observations during the pre and post COVID19-lockdown is poor and simplistic, and is not linked to the airborne data obtained with the MOASA platform.

**Specific comments**

Abstract: The abstract does not include any findings. Please highlight them.

Introduction: The introduction is too long and has rather the character of a measurement report than of a scientific work. Most of the details and acronyms are neither relevant for the data analysis presented or further mentioned in the manuscript. Please move them to the supplement. In particular, the information about COVID19 can be reduced to a paragraph and does not require a separate section.

Line 94. Please include a more specific publication for GOME than Molina and Molina, 2004.

Line 122: Please specify what is meant by "*an introduction to the vertical structure of pollutants during COVID 19 period*". Which are the findings? Please include them in the conclusions and the abstract.

Line 145 Section 2: This section should be shortened and most of the information moved to the supplement. Please include in the main text a table with a summary of the instruments and their most important features, sensitivity, detection limit etc and refer to the supplement for details for the measurement techniques, calibration procedures, flagging of data, etc.

Line 167: It is confusing to mention an Appendix A that actually is a Figure A1. This happens with the rest of appendices all over the text, which are difficult to be identified. Please include them in the supplement and name them accordingly (e.g. Figure S1 in the supplementary information).

Line 203: Why is specifically mentioned that the instrument measures up to 3000 ppbv? This $NO_2$ mixing ratio is not expected to be frequent in airborne measurements in the BL if not directly flying inside industrial plumes. How is the accuracy at the lower end and detection limit? Please clarify also in relation to the sensitivity given in Line 242.

Line 234: Figure 1 and Figure 2 do not seem to be mentioned in the text. Please correct.

Line 241: It would be more informative to show the data of this separate experiment instead of the data that were corrupted by the impact of the filter in the pump performance.

Line 259: Does it mean that the first 7 flights do not have any valid $O_3$ data?

Line 335: Please revise the figure caption of the Appendix B, i.e., figure B1 to make it more understandable: "*The vertical bars represent the error in response for each bead size and is the mean standard error of the mean for 15 second segments of each bead response.*"

Line 359: Is Appendix 5 the in Line 369 mentioned Appendix C, which in reality are the Figures C1 and C2? Please clarify and come to a systematic naming of the so-called appendices to avoid confusion.

Line 370: "… *it is subject to potential uncertainties*…". Are these uncertainties the size distribution uncertainties described in the following section or additional uncertainties for other reasons? Please clarify in the text and revise the necessity of a separate 2.7.3 section for this. As recommended above, part of these details should anyway move to the supplement.

Line 415: Is the Appendix D the table D1?

Line 422 "were" instead of "are"

Line 431: Why is Figure 8 mentioned before Figure 7? Please change the numbering.

Line 444 to 505: All these subsections are not necessary and can be removed. To the main manuscript belongs the content of the summary (3.7) and the figures 7-8. The table 1 and the rest of the information should move to the supplement.

Line 517: I guess that Appendix 7 is the Table F1. See comments above related to the naming and numbering of Appendices.

Line 554: It is not clear how the so-called "ground distances" are classified. What is the reference used? Are they distances over the same geographical area during different flights, distances to a selected source, distances flown during the same time interval at different velocities of the aircraft? Please clarify.

Line 558: What is the interest and usefulness of having a flight mean variability in the case of different tracks or transects over areas of different chemical and meteorological complexity? In a rough analysis, without getting into the details of the sampling area, it is somehow obvious that in a non-remote atmosphere, the greater the sampling scale the more difficult becomes to observe the effect of individual pollution sources. The suitability of the scale will depend on the characteristics and pollution complexity of the area studied.

Line 580: "*For $NO_2$ this absolute variability is below 7.35 ppbv and for particulate counts below 2412.830 counts/second for 90% of data points*" What are the implications of these results for the analysis of the area sampled when using the regional model?

Line 592: "... *no statistical post processing has been applied*". How will this affect the results obtained? Please clarify what you expect.

Line 598: From here the structure of the section 4.3 is a bit erratic: the subsection of $O_3$ mentions both flights but only discusses one, the subsection of $NO_2$ and the BL height discusses only M270 and the following section seems to focus on the $NO_2$ of the M296. I would recommend to discuss the flights individually and to summarise the findings at the end.

Line 604: In Flight M270 the model data seem not to reproduce the variability of the observations at any altitude (the model results vary in each level a few ppb while the observations vary around 20 ppb). Do you have any reason for this? The M296 shows quite a different pattern. Please discuss these differences.

Line 609: "*It may be possible to use the aircraft observations to help identify sources....*" It would be very useful to see if any correction based on the MOASA data (such of this used by Savage et al. ) introduces any improvement in the case studies presented. Similarly, it should be shown the effect of replacing the modelled by the observed BL height.

Line 613: "...shows "the" comparison"

Line 615: Actually, the largest difference between model and observations seems to be below 600 m (in red). Please change the way of plotting the difference (a colour scale is not clear enough) and explain the differences more accurately.

Line 617: How solid is this interpretation? Has this pattern been observed on other days? Is the whole M296 within the observed BL?

Line 620: Why the error in the altitude of the BL can lead to any conclusion about the agreement at the surface (within the BL of the model)?

Line 625: Please change the colour scale for the difference plot. It is impossible to see any difference by this large range.

Line 627: It would be informative to see the Circuits 2 and 3 in the supplement. Please include them.

Line 630: There is about 1 h difference between the first and the last circuit. It is realistic to talk about the same plume? As stated in Line 633 the plume aloft has greater $NO_2$ concentrations. How do you explain this if both circuits are within the BL?

Line 636: How is the comparison between $O_3$ modelled and observed in those circuits?

Line 640: As the concentrations at the ground level are also as low as in the model and so different from the airborne measurements, these results indicate that you have a real gradient in the pollutant concentrations within 423 m (altitude of the first circuit), that I guess is still within the BL on that day. Please comment on this. Is that gradient also visible in other species measured during this flight?

Line 646-647: Is that the case for M296?

Line 651: Please change the scale of $NO_2$ in Figure 16 to enable a more accurate comparison with the airborne data (should not be larger than 50 mg/m$^3$). The range of the London AURN data can be specified in the figure caption. Generally, a table with the values

used for the comparison would facilitate the interpretation. In the figure caption please correct "corresonding"

Line 654: I do not understand what is the meaning of 6.6 or 4.6 sampling sites. How can you have a fraction of a sampling site? Please clarify.

Line 663: Please rephrase. "reform" does seem to indicate a null cycle

Line 664: "*As such, the increase in $O_3$ is coincident with a reduction of the observed $NO_2$ aloft, which, in addition to being reduced by chemical reaction, is also further away from sources (fossil fuel burning, traffic (Jones et al., 2021, Lee et al., 2020)).*" It is not clear what this sentence tries to say. It seems to be quite a simplistic analysis. Please clarify.

Line 666: "*Here, the impact of external factors (meteorology, boundary layer height, seasonal changes, complex chemistry) are not discussed and is beyond the scope of this paper. However, the persistent difference between the surface-based observations and airborne observations aloft demonstrates the importance in quantifying the vertical structure of pollutants…..*" This is quite well known, what is then the scope of this work?

Line 679: If the Appendix E is the table E, the difference between the pre and post COVID19-lockdown averages is well within the standard deviation of the averages in all the sites. That implies that taking into account the large variability of these hourly averages they are not significantly different.

Line 681 to 690: The ozone production is known to be a non-linear and complex process and is not surprising that changes during the lock down cannot be explained by a simple comparison of $NO_2$ and $O_3$ hourly averages of ground-based measurements. As cited in the paper there are a multitude of studies on this subject in the literature. It is not clear why this manuscript includes here such a simplistic interpretation of the data from ground-based stations and then recommend others to make further work in interpreting the data. As these statements do not seem to complement in any form the cited comprehensive analysis of COVID data published by Lee et al, 2020, I recommend either discussing more in detail the relation between ground based and airborne measurements or otherwise removing this part from the manuscript.

Line 723: "*Specifically, we show lower concentrations of $NO_2$ and higher concentrations of $O_3$ aloft.*" Please revise this statement; it does not reflect accurately the results shown.

Line 725: "*Analysis of long-term surface-level trends in the Greater London region show a decrease in NO2 and an increase in $O_3$ following the mandated COVID-19 restrictions*". Please revise carefully the accuracy of this statement based on the analysis and the interpretation presented in the manuscript.

Line 838: Please correct the title of the reference

---

## Referee Comment (RC2)

**Long-term airborne measurements of pollutants over the UK, including during the COVID-19 pandemic, to support air quality model development and evaluation
by Mynard et al.**

**General comments:**

This manuscript describes long-term measurements in the UK of a variety of atmospheric trace species carried out with a Cessna-421 aircraft from June 2019 to April 2022 (including the COVID-19 impacted period). The measurements mainly covered altitudes between the ground and the top of the boundary layer. Specific flight patterns, repeated sorties, were flown sampling both rural and urban regional background pollution in addition to polluted city plumes. The objective was to sample an extended data set to be used for evaluations of the regional air quality model AQUM of the UK Met Office. Here the gained airborne data set is described and first intercomparisons with the model are presented.

The paper addresses scientific questions relevant to the scope of AMT (technical description of instrumentation, novel measurement, modelling).

The manuscript is well-written and generally logical structured. Manuscript is written by native speakers, language fluent and precise, no improvements needed. The number of figures and tables in comparison to the text is appropriate. In general, the figures are of good quality. Proper credit is not always given to related work. The conclusions and statements are rather sparse, based on first results from two case studies. Details on suggestions for improvements on these topics are given in the specific comments.

*For the reasons mentioned above and below the paper is appropriate for publication in AMT after a major revision described below.*

**Specific comments:**

The introductory section is too long and should be improved. It contains detailed descriptions of the AQUM model and the Automatic Urban and Rural Network (AURN), an automatic ground monitoring network, which should be moved to Sect. 2. Describe in Sect. 2 also how vertical mixing is implemented into the model, since this seems to be a crucial parameter for the intercomparison with airborne measurements. To improve the large disagreements between the model and the measurements, it is recommended to implement other schemes to test the influence of the vertical mixing.

In addition, the introductory section contains only few references to previous studies on this topic, e.g. Savage et al. (2013). Include more of such studies and results (as given in the Savage paragraph) and instead shorten some general information at the beginning.

In Sect. 1.1 (Impact of COVID-19) incorporate and discuss results from other studies in Europe related to O3 and NO2 during COVID-19.

In Sect. 2.1 (Instrumentation – general setup) add a table listing the instrumentation, technique, precision, and references.

The results of the study (model evaluation) are rather sparse described on only 3 pages (page 16-19) compared the rather extended manuscript. Include a few more intercomparisons focusing on problems in the model (e.g. boundary layer height, vertical mixing). Add some

more examples from other flights comparing the modelled and measured BL height. Discuss ways to improve the BL height in the model (add a new section "Discussion" ahead of the "Conclusions"). What about the influence of inversion layers located below the BL? Have such cases been observed in the winter flights and how does the model behave?

On Page 19 (line 676-678) you write: "We define the pre-lockdown period as 26th March 2018 to 25th March 2020 and the post-lockdown period as 26th March 2020 to 25th March 2022, where comparing like-for-like months pre- and post- lockdown minimises the impact of seasonality on the comparison." → would it not make more sense to define three periods (pre-lockdown, main lockdown, post-lockdown?

In general, it is recommended to give mean NO2 and O3 values for all flights in a table for a better overview of the airborne results.

After Sect. 4 and before Sect. 5, a section on discussion of the results is missing.

**Minor comments and technical corrections:**

Page 10, line 359:
Appendix 5 not available.

Page 12, line 413:
gcm3 → g/cm3

Page 14, line 506-508:
"Datasets obtained during the MOASA Clean Air project are openly available from the Centre for Environmental Data Archive (CEDA) "Collection of airborne atmospheric measurements for the MOASA Clean Air project" repository (DOI: 10.5285/0aa1ec0cf18e4065bdae8ae39260fe7d)."
   → Add this also at the end of the manuscript at the appropriate place.

Page 14, line 517:
Appendix 5 not available.

Page 15, line 530:
Add from who.

Page 15, line 547:
Why data only used until July 2021 (44 flights) and not all available data until April 2022 (63 flights)?

Page 16, line 582:
Sect. 4.2 is missing.

Page 17, line 607:
"Savage et al. (2013) also reported biases during a ground-site AQUM comparison." → discuss in the manuscript

Page 17, line 633:

"The AQUM model shows little variation and comparatively low NO2 concentration in all circuits above the city" → This result gives little confidence in the model. Can you add other examples, where a city plume is well simulated by the model? Discuss the reasons for the differences.

Page 20, line 732:
Add from who.

Page 21-22:
Some of the references are too sparse. Add more information how to find them:

- Air Quality Expert Group: Fine Particulate Matter ( PM 2 . 5 ) in the United Kingdom, 2012

- DEFRA: Clean air strategy 2019., 2019

- Ecotech: Aurora3000 Integrating Nephelometer with backscatter, 2009.

Page 25:
Give Appendix text first after Figs. 1-16.

Page 26, line 986:
Subtitle "Case study" somehow misplaced.

Page 29, Fig. 3:
Legend covers data, move it.
Blue straight line better visible in red.

Page 32, Fig. 8:
Here only 45 flights shown, why not all 63 flights shown as described in the abstract?

Page 34, Fig. 11:
In the figure text "0.85" is mentioned twice.

Page 43, line 1163:
gcm3 → g/cm3

Page 43, Table E1:
Add NO2 to the header of column 3 and O3 to the header of column 4.

---

## Author Comment (AC1)

We would like to thank reviewer #1 for taking the time to review this manuscript and for providing valuable feedback and suggestions that helped us to significantly improve the manuscript. We have carefully considered all the comments and revised the manuscript accordingly. In particular, the authors are grateful for the reviewers' comments on section 4.4, which has subsequently been removed and replaced with a new section, "Ground-based and airborne observation comparison using long term observations over London".

Below are reviewer #1's Specific Comments, in black, with an in-line corresponding reply from the authors in blue. Where multiple questions are asked in one comment, the author response is bulleted. Some revised text has been omitted due to length.

**Referee Specific comments:**

1.  Abstract: The abstract does not include any findings. Please highlight them. Findings have been highlighted as follows**: "**_These case studies show that for observations of relative humidity, nitrogen dioxide and particle counts, natural pollutant variability is well observed by the aircraft, whereas SO$_2$ variability is limited by instrument precision. Good agreement is seen between observations aloft and those on the ground, particularly for PM$_{2.5}$. (r$^2$ = 0.90). Analysis of odd oxygen suggests titration of ozone is the dominant chemical process throughout the column for the flights analysed, although a slight enhancement of ozone aloft is seen. Finally, a preliminary evaluation of AQUM performance for two case-studies suggests a large positive model bias for ozone aloft, coincident with a negative model bias for NO$_2$ aloft. On one case, there is evidence that an under prediction in the modelled boundary layer height contributes to the observed biases at elevated altitudes._"

2.  Introduction: The introduction is too long and has rather the character of a measurement report than of a scientific work. Most of the details and acronyms are neither relevant for the data analysis presented or further mentioned in the manuscript. Please move them to the supplement. In particular, the information about COVID19 can be reduced to a paragraph and does not require a separate section.

*   This paper introduces the MOASA measurement platform, flight strategies and instrumentation and is not intended to be an in-depth diagnostic analysis, but rather a comprehensive technical reference for future users of these data, including illustrations of the potential uses of these upper air observations for regional-scale model evaluation. This has been emphasised in the abstract and main manuscript. The more detailed analyses to which the referee refers in "General Comments" is intended to follow in future work by both the authors and other database users.

*   The use of acronyms has been reviewed throughout and revised as necessary.

*   The authors agree with the reviewer's comments on the applicability of the COVID discussion throughout the manuscript. The description and subsequent analysis based on the COVID lockdown has been removed, as has mention of COVID in the title. The original section 4.4 has been replaced with a comparison of airborne vs ground-based observation of pollutants over greater London. The only remaining COVID reference is informing the reader that the observation period encompasses the COVID-affected period.

3. Line 94. Please include a more specific publication for GOME than Molina and Molina, 2004. *This reference has been replaced with Liu, X., Chance, K., Sioris, C. E., Spurr, R. J. D., Kurosu, T. P., Martin, R. V. and Newchurch, M. J.: Ozone profile and tropospheric ozone retrievals from the Global Ozone Monitoring Experiment: Algorithm description and validation, J. Geophys. Res. Atmos., 110(20), 1–19, doi:10.1029/2005JD006240, 2005.*

4. Line 122: Please specify what is meant by "*an introduction to the vertical structure of pollutants during COVID 19 period*". Which are the findings? Please include them in the conclusions and the abstract. Please refer to author response to comment #2.

5. Line 145 Section 2: This section should be shortened and most of the information moved to the supplement. Please include in the main text a table with a summary of the instruments and their most important features, sensitivity, detection limit etc and refer to the supplement for details for the measurement techniques, calibration procedures, flagging of data, etc. Please refer to author response to comment #2. This section is highly applicable to the manuscript and thus remains. The authors thank the reviewer for the suggestion of a summary table which has been added.

6. Line 167: It is confusing to mention an Appendix A that actually is a Figure A1. This happens with the rest of appendices all over the text, which are difficult to be identified. Please include them in the supplement and name them accordingly (e.g. Figure S1 in the supplementary information). Appendices have been reviewed and revised throughout.

7. Line 242: Why is specifically mentioned that the instrument measures up to 3000 ppbv? This NO2 mixing ratio is not expected to be frequent in airborne measurements in the BL if not directly flying inside industrial plumes. How is the accuracy at the lower end and detection limit? Please clarify also in relation to the sensitivity given in Line 242. The measurement ceiling has been removed and the sensitivity in line 242 has been clarified as follows: *"The sensitivity of the $NO_2CAPS$ was empirically derived to be 0.17 ± 0.14σ ppbv (during a separate ground-based zero test, where data is also averaged over 10s intervals)."*

8. Line 234: Figure 1 and Figure 2 do not seem to be mentioned in the text. Please correct. Mention of fig 1 and fig 2 have been added to sec. 2 and 2.1, respectively.

9. Line 241: It would be more informative to show the data of this separate experiment instead of the data that were corrupted by the impact of the filter in the pump performance. The authors feel figure 3 is informative to the baseline correction discussed in the main text. However, based on this feedback, the figure has been revised and figure text has been updated accordingly, as shown below.

[Figure]

*Figure 3: Top: timeseries of raw (uncorrected) and processed (corrected) NO₂ concentration. Oscillations seen in the raw and processed data during the filter test are an artefact of the filter, which impacted performance of the instrument pump. These oscillations have been minimised by arbitrarily smoothing (60 second rolling) the data, for visualisation purposes only. Bottom: NO₂ instrument baseline against cell pressure, coloured by altitude, with a linear fit shown as a red line. All data from 11:55:00 to 12:50:00 during flight M304 on 4th November 2021, averaged over 10 second intervals.*

10. Line 259: Does it mean that the first 7 flights do not have any valid O3 data? Yes. The text has been changed to: "…*following some initial issues with negative calculated ozone values during MOASA measurements (impacting the first 7 flights which do not have valid ozone data), the Dewlines were regularly replaced…*".

**11.** Line 335: Please revise the figure caption of the Appendix B, i.e., figure B1 to make it more understandable: "*The vertical bars represent the error in response for each bead size and is the mean standard error of the mean for 15 second segments of each bead response.*" This has been revised as follows: "*Vertical bars show the error in response for each bead size, derived by calculating the standard error of the mean for 15 second segments of each bead response, and then taking the mean of these values.*"

12. Line 359: Is Appendix 5 the in Line 369 mentioned Appendix C, which in reality are the Figures C1 and C2? Please clarify and come to a systematic naming of the so-called appendices to avoid confusion. Appendices have been reviewed and revised throughout.

13. Line 370: "… *it is subject to potential uncertainties*…". Are these uncertainties the size distribution uncertainties described in the following section or additional uncertainties for other reasons? Please clarify in the text and revise the necessity of a separate 2.7.3 section for this. As recommended above, part of these details should anyway move to the supplement.

- Regarding uncertainties, this has been clarified in the text which has been changed to: "….*it is subject to potential uncertainties, such as assumptions of aerosol homogeneity and sphericity, that caution against its use as an accurate measure of the true ambient particle IOR (Frie and Bahreini, 2021).* ".
- Please refer to response to Comment #2 regarding moving essential details to the supplement.

14. Line 415: Is the Appendix D the table D1? Appendices have been reviewed and revised throughout.

15. Line 422 "were" instead of "are" This has been changed.

16. Line 431: Why is Figure 8 mentioned before Figure 7? Please change the numbering. Done.

17. Line 444 to 505: All these subsections are not necessary and can be removed. To the main manuscript belongs the content of the summary (3.7) and the figures 7-8. The table 1 and the rest of the information should move to the supplement. Please refer to response to Comment #2 regarding moving essential details to the supplement.

18. Line 517: I guess that Appendix 7 is the Table F1. See comments above related to the naming and numbering of Appendices. Appendices have been reviewed and revised throughout.

19. Line 554: It is not clear how the so-called "ground distances" are classified. What is the reference used? Are they distances over the same geographical area during different flights, distances to a selected source, distances flown during the same time interval at different velocities of the aircraft? Please clarify. The text has been updated to: "…*Measured values in each dataset were split into groups of equal size, with sizes corresponding to equivalent ground distances ($d_{int}$) ranging from 0.42 km to 17 km, in 0.085 km (1 second) intervals* (*where, for this study which focuses on average variability over a campaign, an airspeed of 85 m/s is assumed to be equivalent to 0.085 km straight-line distance at ground level*)…"

20. Line 558: What is the interest and usefulness of having a flight mean variability in the case of different tracks or transects over areas of different chemical and meteorological complexity? In a rough analysis, without getting into the details of the sampling area, it is somehow obvious that in a non-remote atmosphere, the greater the sampling scale the more difficult becomes to observe the effect of individual pollution sources. The suitability of the scale will depend on the characteristics and pollution complexity of the area studied. We thank the reviewer for their comment. This section has been revised and shortened. The flight mean variability has been removed as the authors feel it did not add to the manuscript. Additional references and text have been added to the section to substantiate the analysis and emphasis the potential use of high-horizontally resolved data to study the natural and often sub-grid variability of pollutants.

21. Line 580: "*For NO2 this absolute variability is below 7.35 ppbv and for particulate counts below 2412.830 counts/second for 90% of data points*" What are the implications of these results for the analysis of the area sampled when using the regional model? Please refer to the previous comment.*"*

22. Line 592: "… *no statistical post processing has been applied*". How will this affect the results obtained? Please clarify what you expect. Changed text to *"… no routine statistical post-processing (SPP, which uses surface level observations to apply corrections to the surface model level only) has been applied to the data. Given this study*

*focuses on those data above the surface level, the omission of the SSP has no impact on the evaluation.*"

23. Line 598: From here the structure of the section 4.3 is a bit erratic: the subsection of $O_3$ mentions both flights but only discusses one, the subsection of NO2 and the BL height discusses only M270 and the following section seems to focus on the NO2 of the M296. I would recommend to discuss the flights individually and to summarise the findings at the end. Agreed – this section has been reformatted as suggested, whereby flights are discussed individually and summarised at the end.

24. Line 604: In Flight M270 the model data seem not to reproduce the variability of the observations at any altitude (the model results vary in each level a few ppb while the observations vary around 20 ppb). Do you have any reason for this? The M296 shows quite a different pattern. Please discuss these differences.
The following has been added to the respective sections:
Flight M270:  *" The variability observed is poorly represented by the coarse resolution model. Variation in the AQUM model data is largely caused by changing from one grid box to the other and ozone shows a typically smooth gradient between model grid boxes. We note that in this case the stacked flight transects only cross a very small number of model cells (3 or 4) in the horizontal, which may be accountable for the low model variability seen here."*
Flight M296: *"…unlike flight M270, the observations and model show similar variability. This is likely due to the flight track crossing a larger number of model cells which encompass more model predictions, and may also be due to the model capturing more variability for this case.."*
And in the summary for this section: *" Variability in modelled ozone appears to be dependent on the number grid boxes encompassed by the flight track. It is expected that ozone concentration in higher resolution models (>12km) will better match variation in the airborne observational data, as model resolution moves towards natural scale variability."*

25. Line 609: *"It may be possible to use the aircraft observations to help identify sources…."* It would be very useful to see if any correction based on the MOASA data (such of this used by Savage et al. ) introduces any improvement in the case studies presented. Similarly, it should be shown the effect of replacing the modelled by the observed BL height. These more in-depth analysis is beyond the scope of the paper and is intended for future work - please refer to author response to Comment #2.

26. Line 613: "…shows "the" comparison" Amended.

27. Line 615: Actually, the largest difference between model and observations seems to be below 600 m (in red). Please change the way of plotting the difference (a colour scale is not clear enough) and explain the differences more accurately. The authors feel the presentation of the data (latitude by altitude) provides good representation of spatial change with altitude. To better highlight the largest difference above 650m, the colour-scale of the difference plot has been revised, the marker size now increases with divergence away from zero, and an additional description has been added to the figure text.

[Figure]

**Figure 17:** *Longitude-altitude plot of NO₂ concentration for vertically stacked transects during flight M270 on 15ᵗʰ September 2020. The left-hand figure shows the aircraft data, the middle figure shows the model data, and the right-hand figure shows the difference between the model and aircraft, where opacity and thickness increase as the difference diverges away from zero. Data averaged over 10 second intervals.*

28. Line 617: How solid is this interpretation? Has this pattern been observed on other days? Is the whole M296 within the observed BL?
- The wording has been changed to: *"…This indicates a potential under-prediction in boundary layer height that may be responsible for the poor predication of NO₂ at elevated altitudes …"*
- The whole of flight M296 is within the observed BL. Analysis of other flights is beyond the scope of this paper but is hoped to be addressed in future work.
29. Line 620: Why the error in the altitude of the BL can lead to any conclusion about the agreement at the surface (within the BL of the model)? The authors believe this comment refers to: "*This comparison indicates the value of evaluating model performance throughout the atmospheric column and suggests that the good agreement of NO2 seen at the surface may in this case have been somewhat fortuitous*.". This section has been revised, restructured, and reworded and – given that (in line with the reviewer's comments) the paper does not strictly include model performance at the surface - it has been removed.
30. Line 625: Please change the colour scale for the difference plot. It is impossible to see any difference by this large range. This whole figure, including the colour scale for the difference plot has been revised as per the below. Please note the corrected ground-based concentrations which are generally in agreement with the airborne data.

[Figure]

**Figure 18:** *Aircraft flight tracks coloured by NO$_2$ concentration (µg/m$^3$) for the first (left, 11:23 to 11:43) and fourth (right, 12:33 to 12:52) circuit, at altitudes of 423 and 657 metres, respectively, around Birmingham during flight M296 on 1$^{st}$ July 2021. Top row shows the aircraft data, middle row shows the model data and bottom row shows the difference between the model and observations. Observation data is from straight and wings level transects and all data is averaged over 10 second intervals. Wind barbs are only shown where the observed wind components exceed the measurement uncertainty. Data in triangles is the hourly surface level AURN NO$_2$ concentration for the circuit. Stars/squares show the location of the Birmingham supersite/airport, respectively Map tiles by Stamen Design, under CC BY 3.0. Data by OpenStreetMap, under ODbL.*

31. Line 627: It would be informative to see the Circuits 2 and 3 in the supplement. Please include them. Circuits 2 and 3 (below) have been added as appendix D, as per below:

[Figure]

**Figure D1:** *Aircraft flight tracks coloured by NO₂ concentration (µg/m³) for the second (left, 11:43:00 to 12:10:00 and third (right, 12:10:00 to 12:33) circuit, at altitudes of 511 and 573 metres, respectively, around Birmingham during flight M296 on 1ˢᵗ July 2021. Top row shows the aircraft data, middle row shows the model data and bottom row shows the difference between the model and observations. Observation data is from straight and wings level transects and all data is averaged over 10 second intervals. Wind barbs are only shown where the observed wind components exceed the measurement uncertainty. Data in triangles is the hourly surface level AURN NO₂ concentration for the circuit. Stars/squares show the location of the Birmingham supersite/airport, respectively. Map tiles by Stamen Design, under CC BY 3.0. Data by OpenStreetMap, under ODbL*

32. Line 630: There is about 1 h difference between the first and the last circuit. It is realistic to talk about the same plume? As stated in Line 633 the plume aloft has greater NO₂ concentrations. How do you explain this if both circuits are within the BL? The following text has been added *"In consonance with AQUM, light north-westerly winds (0 < 5 knots)*

*associated with the high-pressure system are observed in all circuits. These slack winds (equivalent to a maximum velocity south-eastward at 9.26 km per hour) likely pushed the plume (which is seen in the ground data to be present east of the flight track) south-eastward, accounting for the shift in the observed plume with altitude and time (approximately 1 hour between the first and final circuits). The proximity of the plume to Birmingham airport is also of note in run 4."*

33. Line 636: How is the comparison between $O_3$ modelled and observed in those circuits? Refer to response to comment #32. Regarding the revised figure, the following has been added: *"As expected, given that $NO_2$ is photochemically split during the formation of $O_3$, observed $O_3$ aloft (not shown) is inverse to the $NO_2$ observations, and shows a reduction of approx. 20-30 $\mu gm^3$ at the plume locations at all altitudes."*

34. Line 640: As the concentrations at the ground level are also as low as in the model and so different from the airborne measurements, these results indicate that you have a real gradient in the pollutant concentrations within 423 m (altitude of the first circuit), that I guess is still within the BL on that day. Please comment on this. Is that gradient also visible in other species measured during this flight? Please refer to response to comment #32.

35. Line 646-647: Is that the case for M296? Please refer to response to comment #32.

36. Line 651: Please change the scale of NO2 in Figure 16 to enable a more accurate comparison with the airborne data (should not be larger than 50 mg/m3). The range of the London AURN data can be specified in the figure caption. Generally, a table with the values used for the comparison would facilitate the interpretation. In the figure caption please correct "corresonding" Please refer to response to comment #2. This figure, and associated analysis, has been removed.

37. Line 654: I do not understand what is the meaning of 6.6 or 4.6 sampling sites. How can you have a fraction of a sampling site? Please clarify. Please refer to response to comment #2. This figure, and associated analysis, has been removed.

38. Line 663: Please rephrase. "reform" does seem to indicate a null cycle Please refer to response to comment #2. This figure, and associated analysis, has been removed.

39. Line 664: *"As such, the increase in $O_3$ is coincident with a reduction of the observed $NO_2$ aloft, which, in addition to being reduced by chemical reaction, is also further away from sources (fossil fuel burning, traffic (Jones et al., 2021, Lee et al., 2020))."* It is not clear what this sentence tries to say. It seems to be quite a simplistic analysis. Please clarify. Please refer to response to comment #2. This figure, and associated analysis, has been removed.

40. Line 666: *"Here, the impact of external factors (meteorology, boundary layer height, seasonal changes, complex chemistry) are not discussed and is beyond the scope of this paper. However, the persistent difference between the surface-based observations and airborne observations aloft demonstrates the importance in quantifying the vertical structure of pollutants….."* This is quite well known, what is then the scope of this work? Please refer to response to comment #2. This figure, and associated analysis, has been removed.

41. Line 679: If the Appendix E is the table E, the difference between the pre and post COVID19-lockdown averages is well within the standard deviation of the averages in all the sites. That implies that taking into account the large variability of these hourly averages they are not significantly different. Please refer to response to comment #2. This figure, and associated analysis, has been removed.

42. Line 681 to 690: The ozone production is known to be a non-linear and complex process and is not surprising that changes during the lock down cannot be explained by a simple comparison of NO2 and O3 hourly averages of ground-based measurements. As cited in the paper there are a multitude of studies on this subject in the literature. It is not clear why this manuscript includes here such a simplistic interpretation of the data from ground-based stations and then recommend others to make further work in interpreting

the data. As these statements do not seem to complement in any form the cited comprehensive analysis of COVID data published by Lee et al, 2020, I recommend either discussing more in detail the relation between ground based and airborne measurements or otherwise removing this part from the manuscript. Please refer to response to comment #2. This figure, and associated analysis, has been removed.

43. Line 723: "*Specifically, we show lower concentrations of $NO_2$ and higher concentrations of $O_3$ aloft.*" Please revise this statement; it does not reflect accurately the results shown. Please refer to response to comment #2. This figure, and associated analysis, has been removed.

44. Line 725: "*Analysis of long-term surface-level trends in the Greater London region show a decrease in NO2 and an increase in $O_3$ following the mandated COVID-19 restrictions*". Please revise carefully the accuracy of this statement based on the analysis and the interpretation presented in the manuscript. Please refer to response to comment #2. This figure, and associated analysis, has been removed.

45. Line 838: Please correct the title of the reference. This reference is no longer included.

---

## Author Comment (AC2)

We would like to thank reviewer #2 for taking the time to review this manuscript and for providing valuable, constructive feedback and suggestions that helped us to further improve the manuscript. We have carefully considered all the comments and revised the manuscript accordingly.

Below are reviewer #2's Specific Comments, in black, with an in-line corresponding reply from the authors in blue.

**Referee Specific comments:**

1.  A- The introductory section is too long and should be improved. It contains detailed descriptions of the AQUM model and the Automatic Urban and Rural Network (AURN), an automatic ground monitoring network, which should be moved to Sect. 2. The authors feel the AQUM description is applicable to the introductory section but has been shortened. The AURN description has been moved to section 4.2, a new section: "Ground-based and airborne observation comparison using long term observations over London" (refer to comment #3).
    B- Describe in Sect. 2 also how vertical mixing is implemented into the model, since this seems to be a crucial parameter for the intercomparison with airborne measurements. Vertical mixing in the AQUM has been summarised in section 1. Given the non-diagnostic intentions of the paper (see following comment), the authors feel the following is sufficient detail: *"There are 8 vertical levels up to a model top height of 39 km and mixing is parameterised throughout the full depth of the troposphere using a non-local, first order closure, multi-regime scheme (Lock et al., 2000)."*
    C- To improve the large disagreements between the model and the measurements, it is recommended to implement other schemes to test the influence of the vertical mixing. This paper introduces the MOASA measurement platform, flight strategies and instrumentation and is not intended to be an in-depth diagnostic analysis, but rather a comprehensive technical reference for future users of these data, including illustrations of the potential uses of these upper air observations for regional-scale model evaluation. Both the abstract and introduction have been amended to emphasise this. Thus implementation of other schemes to test the influence of vertical mixing is beyond the scope of this paper but is hoped to follow in future work.
    D - In addition, the introductory section contains only few references to previous studies on this topic, e.g. Savage et al. (2013). Include more of such studies and results (as given in the Savage paragraph) and instead shorten some general information at the beginning.
    *The paragraph has been amended as follows: "Comparisons of AQUM to AURN observations (Savage et al. (2013), Neal et al., (2017)), found that AQUM generally performed well, in particular for large air quality events, but had a number of systematic biases. For example, a positive bias in ozone at urban sites, a positive/negative nitrogen oxide ($NO_2$) bias at rural/urban sites and small negative biases in $PM_{2.5}$. These findings are generally conducive to similar air quality model evaluations that employ AURN observations, such as Williams et al., 2018 (10 km CMAQ-Urban model) and Neal et al., 2017 (HadGEM3-RA 50 km regional composition-climate model), where the latter showed a small positive bias in modelled $PM_{2.5}$. For AQUM, ground based observations are used to bias-correct the model data and minimise some of these systematic biases at the surface (Neal et al., 2014). These biases have the potential to introduce bias into any future predictions (Williams et al., 2018).*

2.  In Sect. 1.1 (Impact of COVID-19) incorporate and discuss results from other studies in Europe related to O3 and NO2 during COVID-19. In order to improve the manuscript (following additional feedback on this section) the description and subsequent analysis

based on the COVID lockdown has been removed and replaced with a more relevant section which compares airborne and ground-based observations over greater London. The only remaining COVID reference is to inform the reader that the observation period encompasses the COVID-affected period.

3. In Sect. 2.1 (Instrumentation – general setup) add a table listing the instrumentation, technique, precision, and references. Agreed, this has been added.

4. The results of the study (model evaluation) are rather sparse described on only 3 pages (page 16-19) compared the rather extended manuscript. Include a few more intercomparisons focusing on problems in the model (e.g. boundary layer height, vertical mixing). Add some more examples from other flights comparing the modelled and measured BL height. Discuss ways to improve the BL height in the model (add a new section "Discussion" ahead of the "Conclusions"). What about the influence of inversion layers located below the BL? Have such cases been observed in the winter flights and how does the model behave? Please refer to author response to comment #1C. These suggestions are somewhat out of scope of this technical-focus paper but are hoped to be address in future work.

5. On Page 19 (line 676-678) you write: "We define the pre-lockdown period as 26th March 2018 to 25th March 2020 and the post-lockdown period as 26th March 2020 to 25th March 2022, where comparing like-for-like months pre- and post- lockdown minimises the impact of seasonality on the comparison." □ would it not make more sense to define three periods (prelockdown, main lockdown, post-lockdown? Please refer to author response to comment #2, which advises that this section has been removed.

6. In general, it is recommended to give mean NO2 and O3 values for all flights in a table for a better overview of the airborne results. Please refer to author response to comment #2.

7. After Sect. 4 and before Sect. 5, a section on discussion of the results is missing. This will be added. A discussion/conclusion has been added to each sub-section of section 4.

**Minor comments and technical corrections:**
8. Page 10, line 359: Appendix 5 not available. Appendices have been reviewed and revised accordingly.

9. Page 12, line 413: gcm3 > g/cm3. Changed.

10. Page 14, line 506-508: "Datasets obtained during the MOASA Clean Air project are openly available from the Centre for Environmental Data Archive (CEDA) "Collection of airborne atmospheric measurements for the MOASA Clean Air project" repository (DOI: 10.5285/0aa1ec0cf18e4065bdae8ae39260fe7d)." Add this also at the end of the manuscript at the appropriate place. Added

11. Page 14, line 517: Appendix 5 not available. Appendices have been reviewed and revised accordingly.

12. Page 15, line 530: Add from who. Added *"…. by contacting the author."*

13. Page 15, line 547: Why data only used until July 2021 (44 flights) and not all available data until April 2022 (63 flights)? The analysis was originally completed whilst the flying campaign was in progress. This analysis has now been revised using all available flights. Figure 11 and the figure description has been updated accordingly.

14. Page 16, line 582: Sect. 4.2 is missing. Well spotted - corrected.

15. Page 17, line 607: "Savage et al. (2013) also reported biases during a ground-site AQUM comparison." discuss in the manuscript. This now refers back to section 1 (see comment 1D).

16. Page 17, line 633: "The AQUM model shows little variation and comparatively low NO2 concentration in all circuits above the city" This result gives little confidence in the model. Can you add other examples, where a city plume is well simulated by the model? Please refer to comment 1C. Other examples are beyond the scope of this non-diagnostic, technically-focused paper.

17. Page 20, line 732: Add from who. Replaced with alternative text indicating new archive.

18. Page 21-22: Some of the references are too sparse. Add more information how to find them:
   - Air Quality Expert Group: Fine Particulate Matter ( PM 2 . 5 ) in the United Kingdom, 2012 Updated
   - DEFRA: Clean air strategy 2019., 2019 Updated
   - Ecotech: Aurora3000 Integrating Nephelometer with backscatter, 2009. Updated
19. Page 25: Give Appendix text first after Figs. 1-16. Done.
20. Page 26, line 986: Subtitle "Case study" somehow misplaced. Amended.
21. Page 29, Fig. 3: Legend covers data, move it. Blue straight line better visible in red. The figure has been updated as follows:

[Figure]

*Figure 3: Top: timeseries of raw (uncorrected) and processed (corrected) NO$_2$ concentration. Oscillations seen in the raw and processed data during the filter test are an artefact of the filter, which impacted performance of the instrument pump. These oscillations have been minimised by arbitrarily smoothing (60 second rolling) the data, for visualisation purposes only. Bottom: baseline against cell pressure, coloured by altitude, with a linear fit shown as a red line. All data from 11:55:00 to 12:50:00 during flight M304 on 4th November 2021, averaged over 10 second intervals.*

22. Page 32, Fig. 8: Here only 45 flights shown, why not all 63 flights shown as described in the abstract? This figure and associated text has been updated to include all flights.

[Figure]

23. Page 34, Fig. 11: In the figure text "0.85" is mentioned twice. Amended.
24. Page 43, line 1163: gcm3 >  g/cm3. Amended.
25. Page 43, Table E1: Add NO2 to the header of column 3 and O3 to the header of column 4. Please refer to author response to comment #3, which advises that this section has been removed.

---

## Referee Report (RR1)

**Referee comment on AMT-2023-15- second version**

The revised manuscript now entitled "Long-term airborne measurements of pollutants over the UK, to support air quality model development and evaluation" by Angela Mynard et al., has been improved. The authors have addressed many of the concerns outlined in my original review but the structure and the content are partly still not satisfying.

Concerning the structure, the authors are reluctant to reduce the size of the introduction and the number of subsections. In particular in section 3, the titles of these subsections do not follow a clear logical central idea. Please replace the name of the section "Flight planning" by a more general one so that it can cover the content of the following subsections, in particular of the 3.8.

The authors emphasise in their answers that a deeper discussion of results is out of scope of this paper which has mainly a technical character. In any case, the introduction of short summary sections now and then with the same title "summary" (3.7, 4.2.4. and 4.3.3) does not help. This might be a suitable format for a scientific talk but it is confusing for a manuscript. In order to gain in concision and clarity I recommend the authors to include the discussion and summary of results in 4.2.4 and 4.3.3 in the section 5 at the end, which should be renamed as "Results and conclusions" or "Summary and conclusions". The few sentences in the 3.7 section do not deserve a summary subsection and should be included in the introductory text just before 3.1.

There are still inconsistencies in the Appendices:

- Line 354 *" (see case study in Appendix C)"*. It actually seems to be Appendix B. The equations inside are by the way still numbered as C1, C2 etc

- Line 400 *" This value is derived by weight-averaging the densities of PM2.5 aerosol components measured during a range of UK field experiments, as detailed in appendix C"*. This seems to be correct as Appendix C.

Concerning the content of the revised manuscript, a critical issue is the new interpretation of some of the results made on the basis of the titration of $O_3$. Titration has a very clear meaning in chemistry and the revised text seems to imply that $O_3$ is titrated by $NO_2$, what is impossible as $O_3$ and $NO_2$ do not react. Be aware that the sum of $O_3$ and $NO_2$ is not NO and a potential reaction of $O_3$ and $NO_2$ does not lead to NO. Please clarify and/or correct. The basic principle of the $O_3$ formation and the relation with $NO_2$ is confusing and/or chemically wrong in:

- Line 633: "*Assuming the simplest chemical setup, whereby chemistry in the vertical is controlled by $O_3$ titration ($O_3 + NO_2 => NO$)*".

- Figure caption of Figure 13: "*odd oxygen ($O_3 + NO_2 = NO$)*"

- Line 703: "*As expected, given that $NO_2$ is photochemically split during the formation of $O_3$, observed $O_3$ aloft (not shown) is inverse to the $NO_2$ observations,*" Here would be by the way very informative to see the $O_3$ concentrations this statement refers to.

- Line 754: "*Comparison of odd oxygen implies that ozone titration is the dominant chemical process throughout the atmosphere and helps explicate the complex vertical structures of $O_3$ and $NO_2$ observed throughout the column.*" In particular, revise thoughtfully the scientific part of this statement, which seems to be wrong and difficult to see on the data shown. What is the meaning of "complex vertical structures" and

how are they explained with a simple titration? How can you justify the statement that "*ozone titration is the dominant chemical process throughout the atmosphere*"?

Finally, the units of concentration and density are systematically wrong all over the text (e.g. such as $\mu$g m$^3$ or g cm$^3$ instead of $\mu$g m$^{-3}$ and g cm$^{-3}$). This can be a too recurrent typo or a conceptual mistake. Please revise carefully the text. Other typos will be probably corrected by the editorial office.

**Minor comments**:

- Line 584: "*Here, the HIL AURN site, observed at 84 µgm3 (fig 11 left: grey 585 square and right: red triangle) is significantly higher than both other ground-sites in the region and the range of (…)*" Do you mean : "Here, the 84 µgm$^{-3}$ NO$_2$ observed at HIL AURN site, (fig 11 left: grey 585 square and right: red triangle) is significantly higher than (…)"?

- Line 674 Please remove " *who, as discussed in sec.1, reported positive model ozone biases during a ground site AQUM comparison*" It is redundant and makes the sentence unnecessarily long.

- Figure caption of Figure 13 is not complete and ends with: "is shown as a". Please complete. What is the meaning of a 1-2-1 line?

- Line 738: "Conclusion and future plans" Please remove "future plans" from the title since they are not evident in the text.

---

## Referee Report (RR2)

**Long-term airborne measurements of pollutants over the UK, to support air quality model development and evaluation
by Mynard et al.**

**Technical corrections:**

Page 12, line 442:
gcm3 → gcm-3

Page 54, Table C1 header:
gcm3 → gcm-3

Page 55, line 340:
gcm3 → gcm-3

Page 23, line 853:
You write: "Data is openly available."
However, no link is given. Add data link.

---

## Author Response (AR2)

Many thanks again to reviewer #1 for taking the time to review the revised manuscript and provide valuable feedback that helped us to further improve the manuscript.

Below are reviewer #1's comments, in black, with an in-line corresponding reply from the authors in blue.

1. Please replace the name of the section "Flight planning" by a more general one so that it can cover the content of the following subsections, in particular of the 3.8.
   Sections have been renamed as follows (section number in brackets):
   "*MOASA capability*" >> "*Measurement capability*" (2)
   "*Instrumentation – general setup*" >> "*Instrument overview*" (2.1)
   "*Flight planning*" >> "*Observation and data strategy*". (3)
   "*The MOASA measurement database*" >> "*The measurement database*" (3.7)
   "*Flight data examples*" >> "*Example case studies*" (4)
   "*Conclusion and future plans*" >> "*Summary and conclusions*" (5)

2. … I recommend the authors to include the discussion and summary of results in 4.2.4 and 4.3.3 in the section 5 at the end, which should be renamed as "Results and conclusions" or "Summary and conclusions". Agreed and appreciate the guidance. Text amended as per recommendation.

3. The few sentences in the 3.7 section do not deserve a summary subsection and should be included in the introductory text just before 3.1. Agreed – text has been moved.

4. Line 354 " *(see case study in Appendix C)*". It actually seems to be Appendix B. The equations inside are by the way still numbered as C1, C2 etc. Thank you – these have now been updated correctly.

5. The basic principle of the $O_3$ formation and the relation with $NO_2$ is confusing and/or chemically wrong in:
   a. Line 633: "*Assuming the simplest chemical setup, whereby chemistry in the vertical is controlled by $O_3$ titration ($O_3 + NO_2 => NO$)*". Corrected to $NO + O_3 => NO_2 + O_2$
   b. Figure caption of Figure 13: "*odd oxygen ($O_3 + NO_2 = NO$)*" The equation has been removed from the figure caption.
   c. Line 703: "*As expected, given that NO2 is photochemically split during the formation of O3, observed O3 aloft (not shown) is inverse to the NO2 observations,*" Here would be by the way very informative to see the O3 concentrations this statement refers to. This has been reworded to "*In contrast, observed $O_3$ aloft (not shown) is inverse to the $NO_2$ observations…*".
   d. Line 754: "*Comparison of odd oxygen implies that ozone titration is the dominant chemical process throughout the atmosphere and helps explicate the complex vertical structures of O3 and NO2 observed throughout the column.*" In particular, revise thoughtfully the scientific part of this statement, which seems to be wrong and difficult to see on the data shown. What is the meaning of "complex vertical structures" and how are they explained with a simple titration? How can you justify the statement that "*ozone titration is the dominant chemical process throughout the atmosphere*"?
   The applicable text in section 4.2.3 has been simplified and clarified, to read:
   "*Assuming the simplest mechanism linking chemistry at the ground to that aloft, whereby NO emitted at the surface reacts with $O_3$ via titration to form $NO_2$ ($NO + O_3 => NO_2 + O_2$ ), odd oxygen ($O_x$, in this case defined as the sum of $O_3$ plus $NO_2$ (Bates and Jacob, 2019)) is expected to be conserved throughout the atmospheric profile. Figure 13 shows a comparison of $O_x$ observed at the surface versus aloft for the London sites which yields a regression model gradient of near 1. These results – noting that this simple model neglects mixing, $O_3$ production,*"

*deposition, and other loss mechanisms - are broadly consistent with chemistry via $O_3$ titration being dominant for the cases observed here and indicate that the airborne air masses were coupled to the surface, conducive to the findings of the $PM_{2.5}$ analysis. An $r^2$ of 0.87 also provides confidence that the observations are comparable, regardless of observation technique employed…"*

And the summary (now in section 5, Summary and Conclusions) reads:
*"….For $NO_2$ and $O_3$, chemical processing in the atmospheric column yields an intricate, poorly correlating relationship between airborne and ground-based observations. In contrast, odd oxygen ($O_x = NO_2 + O_3$) at the ground and aloft strongly agree ($r^2 = 0.87$, gradient = 1), suggesting that, for the cases analysed here, ozone titration played a dominant role in the chemistry of these species throughout the atmospheric column. A slight offset in the regression model indicates $O_3$ is higher aloft, suggesting processes unrepresented by this simple model (recalling the limitations noted in section 4.2.3) may also be present. "*

6. The units of concentration and density are systematically wrong all over the text (e.g. such as □g m3 or g cm3 instead of □g m-3 and g cm-3). Please revise carefully the text. Revised throughout.

7. Line 584: "*Here, the HIL AURN site, observed at 84 µgm3 (fig 11 left: grey 585 square and right: red triangle) is significantly higher than both other ground-sites in the region and the range of (…)*" Do you mean : "Here, the 84 µgm-3 $NO_2$ observed at HIL AURN site, (fig 11 left: grey 585 square and right: red triangle) is significantly higher than (…)"?
Text amended to *"Here, the 84 µgm$^3$ $NO_2$ observed at the HIL AURN site (fig 11 left: grey square and right: red triangle) is significantly higher than both other ground-sites in the region and the airborne data (boxplot whiskers in fig 11 left, and track colour in fig 11, right)."*

8. Line 674 Please remove "*who, as discussed in sec.1, reported positive model ozone biases during a ground site AQUM comparison*" It is redundant and makes the sentence unnecessarily long. Agreed and removed.

9. Figure caption of Figure 13 is not complete and ends with: "is shown as a". Please complete. Done What is the meaning of a 1-2-1 line? The text "*representative of a perfect linear relationship*" has been added to the figure 12 and 13 captions.

10. Line 738: "Conclusion and future plans" Please remove "future plans" from the title since they are not evident in the text. This section has been renamed as "Summary and conclusions" as per #1 above.

**Author response to reviewer #2 on AMT-2023-15**

We would like to thank reviewer #2 for taking the time to review this manuscript and for suggestions that helped us to further improve the manuscript. We have revised the manuscript accordingly. Please see in line responses below, in blue.

**Technical corrections:**
Page 12, line 442:
gcm3 to gcm-3
Amended throughout the document.

Page 54, Table C1 header:
gcm3 to gcm-3
Amended throughout the document.

Page 55, line 340:
gcm3 to gcm-3

Amended throughout the document.

Page 23, line 853:
You write: "Data is openly available."
However, no link is given. Add data link.
Link is available.